# ReFORM: Reflected Flows for On-support Offline RL via Noise Manipulation

**Songyuan Zhang**[†]    **Oswin So**[†]    **H. M. Sabbir Ahmad**[‡]    **Eric Yang Yu**[†]
**Matthew Cleaveland**[*]    **Mitchell Black**[*]    **Chuchu Fan**[†]
[†]MIT    [‡]Boston University    [*]MIT Lincoln Laboratory
[†]`{szhang21,oswinso,eyyu,chuchu}@mit.edu`   [‡]`sabbir92@bu.edu`
[*]`{matthew.cleaveland,mitchell.black}@ll.mit.edu`

## ABSTRACT

Offline reinforcement learning (RL) aims to learn the optimal policy from a fixed dataset generated by behavior policies without additional environment interactions. One common challenge that arises in this setting is the out-of-distribution (OOD) error, which occurs when the policy leaves the training distribution. Prior methods penalize a statistical distance term to keep the policy close to the behavior policy, but this constrains policy improvement and may not completely prevent OOD actions. Another challenge is that the optimal policy distribution can be multimodal and difficult to represent. Recent works apply diffusion or flow policies to address this problem, but it is unclear how to avoid OOD errors while retaining policy expressiveness. We propose `ReFORM`, an offline RL method based on flow policies that enforces the less restrictive *support constraint* by construction. `ReFORM` learns a behavior cloning (BC) flow policy with a bounded source distribution to capture the support of the action distribution, then optimizes a reflected flow that generates bounded noise for the BC flow while keeping the support, to maximize the performance. Across 40 challenging tasks from the OG-Bench benchmark with datasets of varying quality and using a *constant* set of hyperparameters for all tasks, `ReFORM` dominates all baselines with *hand-tuned* hyperparameters on the performance profile curves. [1]

## 1 INTRODUCTION

Offline reinforcement learning (RL) trains an optimal policy from a previously collected dataset without interacting with the environment (Levine et al., 2020). This technique is especially useful in domains where large datasets are already available and environment interactions are expensive and potentially unsafe (Fu et al., 2020). However, there are two major challenges. First, the lack of online exploration makes the distribution shift especially dangerous. That is, for out-of-distribution (OOD) actions not represented in the dataset, the learned $Q$-function can produce overly optimistic estimates that lead the policy astray (Levine et al., 2020). Second, traditional policy classes are typically represented using a unimodal distribution such as a Gaussian (Kumar et al., 2020; Tarasov et al., 2023), whereas more complex offline datasets and tasks can require multimodal action distributions.

Prior works attempt to address the OOD issue by keeping the learned policy close to the behavior policy by regularizing a statistical distance to the behavior policy (Wang et al., 2018; Peng et al., 2019; Mao et al., 2023a; Kumar et al., 2019; Wu et al., 2019). However, selecting a distance measurement along with an appropriate regularization weight can be difficult depending on the task and dataset. Perhaps the most common type of statistical distance used is the Kullback–Leibler (KL) divergence (Wang et al., 2018; Peng et al., 2019; Wu et al., 2019; Jaques et al., 2019; Siegel et al., 2020; Nair et al., 2020; Wang et al., 2020; Kostrikov et al., 2022; Park et al., 2025b), which can avoid the OOD issue but can also be too restrictive and produce an overly conservative policy. For example, if the dataset has low density on the optimal behavior, the KL divergence regularization will encourage the learned policy to be suboptimal. Similar works (Wu et al., 2019; Kumar et al.,

---

[1]Project website: `https://mit-realm.github.io/reform/`

2019) have considered alternative statistical distances such as the Wasserstein and MMD distances that have been shown to improve performance on certain tasks. However, these methods do not completely prevent OOD actions, and the need to choose a regularization weight remains a problem.

To tackle the challenge of multimodal action distributions, recent works have proposed using diffusion policies (Hansen-Estruch et al., 2023a) and flow policies (Park et al., 2025b) to model complex action distributions in the dataset. However, it remains unclear how to address the OOD issue with these highly expressive function classes without hurting their expressivity.

In this work, we propose **Re**flected **F**lows for **O**n-support offline **R**L via noise **M**anipulation (ReFORM), an offline RL method that aims to address both above issues by constraining a flow policy using the less restrictive *support constraint*. Rather than regularizing the learned policy via a statistical distance, we only require the actions produced to stay within the support of the action distribution of the behavior policy. ReFORM learns a behavior cloning (BC) flow policy from the dataset, and additionally learns a reflected flow (Xie et al., 2024) noise generator that manipulates the source distribution of the BC policy within its support. This approach enables us to *realize the support constraint by construction* without regularization, therefore avoiding the need to specify any regularization weights. In other words, our method bypasses the hyperparameter sensitivity issue by having *constant hyperparameters*. To summarize our contributions:

- We propose ReFORM, a two-stage flow policy that realizes the support constraint by construction and avoids the OOD issue without constraining the policy improvement.
- We propose applying reflected flow to generate constrained multimodal noise for the BC flow policy, thereby mitigating OOD errors while maintaining the multimodal policy.
- Extensive experiments on $40$ challenging tasks with datasets of different qualities demonstrate that, with a *constant set of hyperparameters*, ReFORM dominates all baselines using flow policy structures with the *best hand-tuned hyperparameters* on the performance profile curve.

## 2 RELATED WORK

**Distributional shift mitigation in offline RL.** A fundamental challenge of dynamic programming methods in offline RL is the OOD problem, where the learned policy tries to exploit erroneous $Q$-values from extrapolation error (Levine et al., 2020) and generates OOD actions. Consequently, many offline RL methods have proposed to constrain or penalize the statistical distance between the learned policy and the behavior policy, either with an additional loss term or by regressing to the estimated optimal policy, to mitigate this distribution shift issue. Examples include using the maximum mean discrepancy (MMD) distance (Kumar et al., 2019), Wasserstein distance (Wu et al., 2019) and KL divergence (Wang et al., 2018; Peng et al., 2019; Wu et al., 2019; Jaques et al., 2019; Siegel et al., 2020; Nair et al., 2020; Wang et al., 2020; Kostrikov et al., 2022; Park et al., 2025b). One key challenge with these methods is that the amount of regularization is a hyperparameter that needs to be tuned for each task and dataset (Park et al., 2025a;b) and can significantly affect the method's performance. Moreover, as argued by Kumar et al. (2019), constraining the divergence can be too restrictive in cases where optimal actions happen with very low probability under the behavior policy. Another family of methods uses the *support* of the behavior policy, either by regularizing the policy (Kumar et al., 2019; Wu et al., 2022; Mao et al., 2023a; Zhang et al., 2023), or via regularizing the $Q$-function outside the support (Kumar et al., 2020; Lyu et al., 2022; Mao et al., 2023b; Cen et al., 2024). Our work falls in the category of enforcing support constraints on the learned policy. However, instead of approximating the support constraint by a suitably designed regularization term, our method enforces the support constraint *by construction* by optimizing in the BC policy's (bounded) latent space.

**Fine-tuning flow-based models for offline RL.** BC methods using diffusion models (Sohl-Dickstein et al., 2015; Ho et al., 2020; Song et al., 2020) or flow matching (Lipman et al., 2023; Liu et al., 2022; Albergo & Vanden-Eijnden, 2022) have seen increasing use in the control and robotics communities (Chi et al., 2023; Reuss et al., 2023; Pearce et al., 2023; Wang et al., 2023). However, since BC aims to mimic the dataset, its performance is tied to the performance of the behavior policy. To fix this, one can consider fine-tuning the learned flow-based model to maximize a user-supplied reward function. Following the success of fine-tuning flow-based models for image generation (Uehara et al., 2024; Black et al., 2024; Domingo-Enrich et al., 2024), fine-tuning has also been applied to the offline RL setting (Hansen-Estruch et al., 2023a; Chen et al., 2024; Park

et al., 2025b; Ding & Jin, 2024; Zhang et al., 2025). However, almost all fine-tuning methods for offline RL tackle the problem of distribution shift with an additional loss term penalizing statistical distance from the behavior policy, with the weight of this term being a sensitive hyperparameter that needs to be tuned for each task and dataset (Park et al., 2025b).

**Latent space optimization in generative modeling.** Instead of fine-tuning the flow model directly, another line of work considers optimizing the distribution in the latent space, i.e., initial noise, of the generative model. In the context of image generation, methods that optimize the initial noise using either regression (Li et al., 2025; Guo et al., 2024; Zhou et al., 2024; Ahn et al., 2024; Eyring et al., 2025) or RL (Miao et al., 2025) have found success in improving the quality of generated images. In RL, Singh et al. (2021) explored using normalizing flows (Dinh et al., 2016) to improve exploration in *online* RL. Since offline RL was not the focus of their work, they do not restrict the output of their learned latent space policy. Consequently, the policy can output unbounded and potentially OOD samples in the latent space, which is harmful in the offline RL setting. Recently, Zhou et al. (2021) and Wagenmaker et al. (2025) have applied this idea to offline RL for Conditional Variational Autoencoders (Sohn et al., 2015) and diffusion policies, respectively, but they additionally restrict the latent space policy to a fixed action magnitude. Here, the action magnitude roughly controls how likely the latent action is under the behavior policy, playing a similar role to the statistical distance regularization coefficient in existing offline RL works. As we will show in Section 5, the final performance is quite sensitive to this hyperparameter, which varies on different tasks and different datasets. In contrast, our proposed method does not have any such hyperparameters that play a similar role, removing the need for adapting them each time the environment or dataset changes.

## 3 PRELIMINARIES

**Offline RL.** Let $\Delta(\mathcal{X})$ be the set of probability distributions over space $\mathcal{X}$, and denote placeholder variables with gray. A Markov Decision Process (MDP) is defined by a tuple $\mathcal{M} = (\mathcal{S}, \mathcal{A}, r, \rho_0, P, \gamma)$, where $\mathcal{S}$ is the state space, $\mathcal{A} \subseteq \mathbb{R}^d$ is the $d$-dimensional action space, $r(s, a) : \mathcal{S} \times \mathcal{A} \to \mathbb{R}$ is the reward function, $\rho_0 \in \Delta(\mathcal{S})$ is the initial state distribution, $P(s'|s, a) : \mathcal{S} \times \mathcal{A} \to \Delta(\mathcal{S})$ is the transition dynamics, and $\gamma \in [0, 1]$ is the discount factor. Given a dataset of $N$ trajectories $\mathcal{D} = \{\tau_1, \tau_2, \ldots, \tau_N\}$ generated by some *behavior* policy $\pi_\beta(a|s) : \mathcal{S} \to \Delta(\mathcal{A})$, where $\tau_i = (s_0, a_0, s_1, a_1, \ldots, s_{H_i}, a_{H_i})$, the goal of offline RL is to find a policy $\pi_\theta(a|s) : \mathcal{S} \to \Delta(\mathcal{A})$ parameterized by $\theta$ that maximizes the expected discounted return $R(\pi_\theta) = \mathbb{E}_{\tau \sim \rho^{\pi_\theta}(\tau)}[\sum_{h=0}^{H} \gamma^h r(s_h, a_h)]$, where $\rho^{\pi_\theta}(\tau) = \rho_0(s_0)\pi_\theta(a_0|s_0)P(s_1|s_0, a_0) \cdots \pi_\theta(a_H|s_H)$. Note that in the offline RL setting, sampling in the environment with policy $\pi_\theta$ is not allowed.

OOD actions are a key challenge in offline RL (Levine et al., 2020). Many actor-critic methods learn the policy-conditioned state-action value function (i.e., $Q$-function) $Q(s, a) : \mathcal{S} \times \mathcal{A} \to \mathbb{R}$. For a policy $\pi_\theta$, this is defined as

$$Q^{\pi_\theta}(s, a) = \mathbb{E}\left[\sum_{h=0}^{H} \gamma^h r(s_h, a_h) \mid s_0 = s, a_0 = a, a_h \sim \pi_\theta(s_h), \forall h \geq 1\right], \quad (1)$$

corresponding to the expected discounted return obtained by applying action $a$ from state $s$ then following policy $\pi_\theta$. For parameters $\phi$, $Q_\phi$ is commonly learned with fitted $Q$ evaluation using the TD error

$$\mathcal{L}(\phi) = \mathbb{E}_{(s,a,s') \sim \mathcal{D}, a' \sim \pi_\theta(s')}\left[\left(r(s, a) + \gamma Q_{\hat{\phi}}^{\pi_\theta}(s', a') - Q_\phi^{\pi_\theta}(s, a)\right)^2\right], \quad (2)$$

where $Q_{\hat{\phi}}^{\pi_\theta}$ is a target network (e.g., with soft parameters updated by polyak averaging (Polyak & Juditsky, 1992)). However, if the policy $\pi_\theta$ samples OOD actions $a'$, the target $Q_{\hat{\phi}}^{\pi_\theta}$ can produce an erroneous OOD value and cause the learned policy to incorrectly optimize for the OOD value (Levine et al., 2020). To address this issue, many offline RL methods regularize the statistical distance between the learned policy and the behavior policy (e.g., with the KL divergence (Peng et al., 2019; Fujimoto & Gu, 2021; Hansen-Estruch et al., 2023b) or Wasserstein distance (Wu et al., 2019; Park et al., 2025b)), resulting in the following objective for policy improvement:

$$\mathcal{L}(\theta) = \mathbb{E}_{s \sim \mathcal{D}, a \sim \pi_\theta(s)}\left[-Q_\phi^{\pi_\theta}(s, a) + \alpha D(\pi_\theta \| \pi_\beta)\right], \quad (3)$$

where $D(\cdot \| \cdot)$ is some statistical distance, e.g., $D_{\mathrm{KL}}$ for KL divergence or $D_{\mathrm{W2}}$ for the Wasserstein distance. However, this regularized objective introduces an additional hyperparameter $\alpha$ that needs to be *hand-tuned* for each experiment (Park et al., 2025a;b).

**Flow matching.**  Flow matching (Lipman et al., 2023; Liu et al., 2022; Albergo & Vanden-Eijnden, 2022) has recently become an increasingly popular way of training flow-based generative models. Given a target distribution $p(x) \in \Delta(\mathbb{R}^d)$, flow matching learns a time-dependent velocity field $v(t, x)$ that transforms a simple source distribution $q(x)$ (e.g. standard Gaussian $\mathcal{N}(0, I^d)$) at $t = 0$ to the target distribution $p(x)$ at $t = 1$. The resulting flow $\psi(t, x) : [0, 1] \times \mathbb{R}^d \to \mathbb{R}^d$, mapping samples from the source $x \sim q$ to the target $\psi(1, x) \sim p$, is then the solution to the ordinary differential equation (ODE)

$$\frac{d}{dt}\psi(t, x) = v(\psi(t, x)), \quad \psi(0, x) = x. \tag{4}$$

Flow matching is a simple yet powerful technique alternative to denoising diffusion (Ho et al., 2020), capable of generating complex multimodal target distributions.

## 4 METHOD

To solve the problem of OOD actions, at any given state $s$, the chosen action $a$ should be constrained to lie within the *support* $\mathrm{supp}(\pi_\beta(\cdot|s)) := \{a \mid \pi_\beta(a|s) > 0\}$ of the behavior policy $\pi_\beta$. However, constraining common statistical distances, such as the KL divergence or the Wasserstein distance, theoretically leads to problems from the perspective of support constraints [2]. We provide the following theoretical results with all proofs provided in Appendix A.

First, constraining the KL divergence is a sufficient but not necessary condition to enforce support constraints (Kumar et al., 2019; Mao et al., 2023a). Formally, we have the following result:

**Proposition 1.** *Given a state $s \in \mathcal{S}$, for any $\epsilon$ such that $0 \le \epsilon < \infty$, $D_{\mathrm{KL}}(\pi_\theta(\cdot|s) \| \pi_\beta(\cdot|s)) \le \epsilon$ implies $\mathrm{supp}(\pi_\theta(\cdot|s)) \subseteq \mathrm{supp}(\pi_\beta(\cdot|s))$. On the other hand, for any $M > 0$, there exist distributions $\pi_\theta$ and $\pi_\beta$ such that $\mathrm{supp}(\pi_\theta(\cdot|s)) \subseteq \mathrm{supp}(\pi_\beta(\cdot|s))$ but $D_{\mathrm{KL}}(\pi_\theta(\cdot|s) \| \pi_\beta(\cdot|s)) > M$.*

Proposition 1 tells us that the KL divergence constraint is more restrictive than the support constraint. This additional restriction has been found to impede the performance improvement of $\pi_\theta$ over $\pi_\beta$ (Mao et al., 2023a). While this issue can be alleviated with a small $\alpha$ in (3), in practice, this can result in OOD problems due to estimation errors (Levine et al., 2020).

Wasserstein distance is another statistical distance used by previous works. However, constraining the Wasserstein distance cannot enforce support constraints despite its strong empirical performance in offline RL (Park et al., 2025b). Formally, we have the following result:

**Proposition 2.** *Given a state $s \in \mathcal{S}$, suppose that $\mathrm{supp}(\pi_\beta(\cdot|s)) \ne \mathcal{A}$. Then, for any $\epsilon > 0$, there exists a policy $\pi_\theta$ such that $\mathrm{supp}(\pi_\theta(\cdot|s)) \not\subseteq \mathrm{supp}(\pi_\beta(\cdot|s))$, but $D_{\mathrm{W2}}(\pi_\theta(\cdot|s) \| \pi_\beta(\cdot|s)) \le \epsilon$.*

Motivated by the above theoretical challenges of the KL divergence and Wasserstein distance in addressing the issue of OOD actions, we instead consider the following support-constrained policy optimization problem to tackle this issue directly.

$$\max_\theta \quad R(\pi_\theta) = \mathbb{E}_{\tau \sim \rho^{\pi_\theta}} \left[ \sum_{h=0}^{H} \gamma^h r(s_h, a_h) \right], \tag{5a}$$

$$\text{s.t.} \quad \mathrm{supp}(\pi_\theta(\cdot|s)) \subseteq \mathrm{supp}(\pi_\beta(\cdot|s)), \quad \forall s \in \mathcal{S}. \tag{5b}$$

Unfortunately, enforcing the support constraint (5b) is a challenging problem since (i) accurately estimating the $\mathrm{supp}(\pi_\beta(\cdot|s))$ (Grover et al., 2018), and (ii) enforcing $\mathrm{supp}(\pi_\theta(\cdot|s)) \subseteq \mathrm{supp}(\pi_\beta(\cdot|s))$ given an estimate of $\mathrm{supp}(\pi_\beta(\cdot|s))$ (Zhang et al., 2023), are both nontrivial to solve for.

To tackle these problems, we propose learning a BC flow policy $\psi_{\theta_1}(t, z; s)$ that transforms a source distribution $q_{\mathrm{BC}}$ into a state-conditioned target distribution $p_{\mathrm{BC}}(a|s) \approx \pi_\beta(a|s)$. In particular, we

---

[2]by interpreting the constant as a Lagrange multiplier, regularization with a fixed coefficient as in (3) can be interpreted as equivalently enforcing a constraint (Levine et al., 2020)

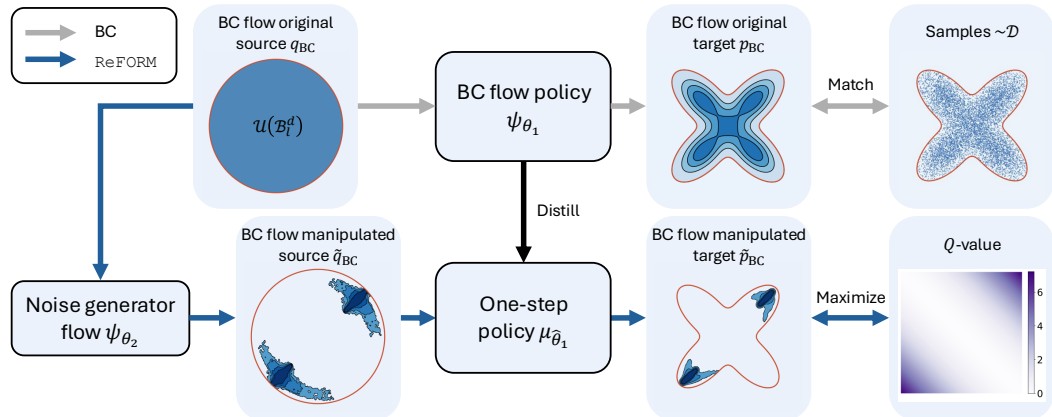

Figure 1: **ReFORM algorithm.** The process with gray arrows indicates the BC flow policy, learned to transform a simple source distribution $q_{\mathrm{BC}} = \mathcal{U}(\mathcal{B}_l^d)$ to a target distribution $p_{\mathrm{BC}}$ that matches the dataset $\mathcal{D}$. The blue arrows indicate the ReFORM process, where we learn a flow noise generator to generate a manipulated source distribution $\tilde{q}_{\mathrm{BC}}$ for the BC policy so that the manipulated target $\tilde{p}_{\mathrm{BC}}$ maximizes the $Q$ value while staying inside the support (denoted in red) of the BC policy.

use a $q_{\mathrm{BC}}$ with bounded support such that $\mathrm{supp}(\pi_\beta)$ can be approximated by the image of $\mathrm{supp}(q_{\mathrm{BC}})$ under the BC flow. One benefit of this approach is that this enables learning a policy that satisfies the support constraints *by construction* by taking advantage of the property that for *any* sample $z \in \mathrm{supp}(q_{\mathrm{BC}})$ within the (bounded) source distribution's support, $\psi_{\theta_1}(1, z; s) \in \mathrm{supp}(p_{\mathrm{BC}}(\cdot|s)) \approx \mathrm{supp}(\pi_\beta(\cdot|s))$. Hence, we propose to construct the policy $\pi_\theta$ as the composition of some *noise generator* with the BC flow $\psi_{\theta_1}$. If the generated noise distribution $\tilde{q}_{\mathrm{BC}}$ has the same support as $q_{\mathrm{BC}}$, i.e.,

$$\mathrm{supp}(\tilde{q}_{\mathrm{BC}}) \subseteq \mathrm{supp}(q_{\mathrm{BC}}) \tag{6}$$

then the pushforward of $\tilde{q}_{\mathrm{BC}}$ under $\psi_{\theta_1}$ naturally satisfies the support constraints (5b). With the support constraint (5b) satisfied by construction, solving the support-constrained policy optimization problem (5) reduces to performing unconstrained optimization of the objective (5a).

**Remark 1.** *This idea of outputting noise is not new. Prior works have proposed similar "noise manipulation/steering" techniques for fine-tuning diffusion models and flow models (Li et al., 2025; Guo et al., 2024; Miao et al., 2025; Wagenmaker et al., 2025). One **key** difference is that we choose the source distribution of the flow model to be a distribution with **bounded** support, which enables better approximation of the support of $\pi_\beta$. Moreover, we propose a different form of the noise generator $\tilde{q}_{\mathrm{BC}}$ than prior works that maintains the high expressivity of flow-based policies.*

We call our method ReFORM, which we summarize in Figure 1. In the following subsections, we elaborate on each of these components in detail.

### 4.1 FLOW-BASED BEHAVIOR POLICY LEARNING

ReFORM begins by learning a BC flow policy that transforms the source distribution $q_{\mathrm{BC}}$ to $p_{\mathrm{BC}}(\cdot|s)$, which approximates $\pi_\beta(\cdot|s)$. We choose $q_{\mathrm{BC}} = \mathcal{U}(\mathcal{B}_l^d)$, the uniform distribution over the $d$-dimensional hypersphere with radius $l$, so that

$$\mathrm{supp}(q_{\mathrm{BC}}) = \mathcal{B}_l^d := \{z \in \mathbb{R}^d \mid \|z\| \le l\}. \tag{7}$$

We discuss the choice of $l$ in Appendix C.4. To learn the BC flow policy $\psi_{\theta_1}$, we learn its corresponding velocity field $v_{\theta_1}(t, z; s) : [0, 1] \times \mathcal{B}_l^d \times \mathcal{S} \to \mathbb{R}^d$ parameterized by $\theta_1$ such that solving the ODE (4) gives actions $a = \psi_{\theta_1}(1, z; s)$ for $z \sim q_{\mathrm{BC}}$. We apply a simple linear flow for learning the velocity field following Park et al. (2025b) with loss

$$\mathcal{L}_{\mathrm{BC}}(\theta_1) = \mathbb{E}_{(s,a)\sim\mathcal{D}, z\sim\mathcal{U}(\mathcal{B}_l^d), t\sim\mathcal{U}[0,1]} \left[ \left\| v_{\theta_1}(t, x_t; s) - (a - z) \right\|^2 \right], \tag{8}$$

where $x_t = (1 - t)z + ta$ is the linear conditional probability path.

## 4.2 REFLECTED FLOW-BASED NOISE MANIPULATION

A key component in enforcing the support constraints as proposed above is the use of a noise generator with the same support as the BC flow-policy's source distribution $q_{\text{BC}}$. Prior works that apply similar "noise manipulation" or "noise steering" techniques implement the generated noise $\tilde{q}_{\text{BC}}$ as a truncated Gaussian (e.g., by clipping or squashing with $\tanh$). However, the use of a *unimodal* $\tilde{q}_{\text{BC}}$ severely limits the expressiveness of $\tilde{q}_{\text{BC}}$ and thus also that of the resulting learned policy $\pi_\theta$.

One way to improve the expressiveness is by replacing the Gaussian distribution with a flow-based generative model, as has been done with the actions. We propose to do the same, but to the *noise* instead. Specifically, we choose to use a flow noise generator $\psi_{\theta_2}(t, w; s) : [0, 1] \times \mathcal{B}_l^d \times \mathcal{S} \to \mathcal{B}_l^d$ and denote its associated velocity field as $v_{\theta_2}(t, w; s) : [0, 1] \times \mathcal{B}_l^d \times \mathcal{S} \to \mathbb{R}^d$. However, the support of a flow-based generative model is generally unconstrained, which violates our requirement on the support of $\tilde{q}_{\text{BC}}$ (6). To resolve this, we propose to use a *reflected flow* (Xie et al., 2024), which can be used to guarantee that samples from $\psi_{\theta_2}$ are contained within $\text{supp}(q_{\text{BC}})$ by considering the following *reflected* ODE (Xie et al., 2024) instead of (4):

$$d\psi_{\theta_2}(t, w; s) = v_{\theta_2}(t, \psi_{\theta_2}(t, w; s); s)dt + dL_t, \quad \psi_{\theta_2}(0, w; s) = w, \tag{9}$$

where the reflection term $dL_t$ compensates the outward velocity at $\partial \text{supp}(q_{\text{BC}})$ by pushing the motion back to $\text{supp}(q_{\text{BC}})$ (Xie et al., 2024).

For convenience, let $\mu_{\theta_1}(z; s) = \psi_{\theta_1}(1, z; s)$ and $\mu_{\theta_2}(w; s) = \psi_{\theta_2}(1, w; s)$, and let $\mu_\theta(w; s) = \mu_{\theta_1}(\mu_{\theta_2}(w; s); s)$ denote their composition. We optimize the noise generator $\psi_{\theta_2}$ to maximize the expected $Q$-value of the learned policy $\mu_\theta$ with the following loss

$$\mathcal{L}_{\text{NG}}(\theta_2) = \mathbb{E}_{s \sim \mathcal{D}, w \sim \mathcal{U}(\mathcal{B}_l^d)} \left[ -Q^{\mu_\theta}(s, \mu_{\theta_1}(\mu_{\theta_2}(w; s); s)) \right], \tag{10}$$

noting that the parameters of the BC policy $\theta_1$ stay fixed when optimizing $\theta_2$.

We have yet to specify the reflection term $dL_t$ in (9), as many choices of $dL_t$ constrain the ODE to remain within $\text{supp}(q_{\text{BC}})$. In particular, we wish for the reflection term $dL_t$ to be robust to numerical integration. Fortunately, $\text{supp}(p_{\text{BC}}) = \mathcal{B}_l^d$ being a hypersphere (7) simplifies this design. Consider solving the normal ODE (4) using the popular Euler method:

$$z_{k+1} = z_k + v_{\theta_2}(k\Delta t, w; s)\Delta t, \quad k \in \{0, \ldots, N - 1\}, \quad \psi_{\theta_2}(1, w; s) \leftarrow z_N, \tag{11}$$

where $N$ is the number of integration steps, $\Delta t = \frac{1}{N}$, and $z_0 = w$. For the reflected case (9), we propose modifying the Euler method (11) by performing a projection back into the hypersphere after every Euler step. This gives us the following reflected Euler method

$$z_{k+1} = \mathbf{1}\{\hat{z}_{k+1} \in \mathcal{B}_l^d\}\hat{z}_{k+1} + (1 - \mathbf{1}\{\hat{z}_{k+1} \in \mathcal{B}_l^d\})\left(\hat{z}_{k+1} - \langle v_{\theta_2}(k\Delta t, w; s)\Delta t, n_{k+1}\rangle n_{k+1}\right), \tag{12}$$

where $\hat{z}_{k+1} = z_k + v_{\theta_2}(k\Delta t, w; s)\Delta t$ follows the original Euler step, $n_k = \frac{\hat{z}_k}{\|\hat{z}_k\|}$, and $\langle \cdot, \cdot \rangle$ is the inner product. We then propose to choose $dL_t$ that is defined implicitly by the above procedure. Note that (12) has the same complexity as (11), because (12) only contains one step projection.

For this to be a valid reflected flow, samples $z$ from the proposed reflected Euler method (12) should satisfy the desired support constraints $z \in \text{supp}(q_{\text{BC}}) = \mathcal{B}_l^d$, which we formally state below.

**Theorem 1.** *The target distribution of the noise generator stays within the support of the original source distribution of the BC policy, i.e.,* $\text{supp}(\tilde{q}_{\text{BC}}) \subseteq \text{supp}(q_{\text{BC}})$.

Combining Theorem 1 with the ideas from above then allows us to formally prove that the resulting action distribution stays within the support $\text{supp}(p_{\text{BC}})$ and hence does not result in OOD actions:

**Theorem 2.** *The manipulated target distribution $\tilde{p}_{\text{BC}}$ of the BC flow policy remains within the support of the original BC policy, i.e.,* $\text{supp}(\tilde{p}_{\text{BC}}) \subseteq \text{supp}(p_{\text{BC}})$.

Theorem 2 guarantees that the learned policy provably avoids OOD actions *without any regularization terms*. This avoids the need for costly hyperparameter tuning for each environment and dataset, and also does not impede the potential improvement of the learned policy.

## 4.3 POLICY DISTILLATION

One drawback of our proposed method is that computing the gradient of the actor loss $\nabla_\theta \mathcal{L}_{\text{NG}}$ (10) requires computing the gradient $\nabla_z \mu_{\theta_1}$, which involves a long backpropagation through time

(BPTT) chain since $\mu_{\theta_1}$ is evaluated with Euler integration. To reduce the computational burden, we follow Park et al. (2025b) and distill (Salimans & Ho, 2022; Geng et al., 2023; 2025) the learned BC flow policy by learning a one-step policy $\hat{\mu}_{\hat{\theta}_1}(z;s) : \mathcal{B}_l^d \times \mathcal{S} \to \mathcal{A}$ parameterized by $\hat{\theta}_1$ that directly maps the latent variable $z$ to the action $a$ with the following distillation loss:

$$\mathcal{L}_{\text{Distill}}(\hat{\theta}_1) = \mathbb{E}_{s \sim \mathcal{D}, z \sim \mathcal{U}(\mathcal{B}_l^d)} \left[ \|\mu_{\hat{\theta}_1}(z;s) - \mu_{\theta_1}(z;s)\|^2 \right]. \tag{13}$$

## 5 EXPERIMENTS

In this section, we conduct experiments to answer the following research questions. Additional details for our implementation, environments, and algorithm hyperparameters, and full results with more ablations are provided in Appendix C.

**(Q1):** How does ReFORM perform compared to other offline RL algorithms with flow policies?

**(Q2):** Does ReFORM avoid the OOD issue without limiting the performance improvement?

**(Q3):** Is it necessary for the BC policy's source distribution to have bounded support?

**(Q4):** Is the reflected flow necessary for generating the targeted noise?

**(Q5):** How is our design of the reflection term?

**(Q6):** Is the distillation of the BC flow policy necessary?

### 5.1 SETUP

**Environments.** We evaluate ReFORM and the baselines on 40 tasks from the OGBench offline RL benchmark (Park et al., 2025a) designed in 4 environments, including locomotion tasks and manipulation tasks. We use two kinds of datasets, CLEAN and NOISY. The CLEAN dataset consists of random environment trajectories generated by an expert policy. The NOISY dataset consists of random trajectories generated by a highly suboptimal and noisy policy.

**Baselines.** We compare ReFORM with the state-of-the-art offline RL algorithms with flow policies, including Flow Q-Learning (FQL) (Park et al., 2025b), Implicit Flow Q-Learning (IFQL) (Park et al., 2025b), and Diffusion Steering via RL (DSRL) (Wagenmaker et al., 2025). Since FQL's performance highly depends on the $\alpha$ hyperparameter (Eq. (3)), we consider three variants of FQL: FQL(M) uses the $\alpha^*$ that is *hand-tuned* for each environment using the CLEAN dataset by Park et al. (2025b), FQL(S) uses $\alpha = \alpha^*/10$, and FQL(L) uses $\alpha = 10 \cdot \alpha^*$. IFQL is the flow version of IDQL (Hansen-Estruch et al., 2023b) implemented in Park et al. (2025b). For DSRL, we use the *hand-tuned* noise bound by Wagenmaker et al. (2025). Note that ReFORM uses the *same* hyperparameters across *all* tasks.

**Evaluation Metrics.** We run each algorithm with 3 different seeds for each task and evaluate each converged model on 32 different initial conditions. We define the *normalized score* for each task as the return normalized by the minimum and maximum returns across all algorithms.

### 5.2 MAIN RESULTS

**(Q1): ReFORM achieves the best overall performance with a constant set of hyperparameters.** As recommended by Agarwal et al. (2021), we plot the performance profile over all tasks with different datasets in Figure 2. It is clear that ReFORM achieves the best performance for both the CLEAN and NOISY datasets. For the CLEAN dataset, DSRL and FQL(M) achieve the second and third best respective performance because their hyperparameters are specifically hand-tuned for these environments. However, for the NOISY dataset, the performance of both DSRL and FQL(M) drops significantly, whereas FQL(S) becomes the second-best method behind ReFORM. This highlights the hyperparameter sensitivity of the baseline methods. Moreover, we observe that when the behavior policy performs poorly (i.e., on NOISY), a stronger density-based regularization impedes the ability of the learned policy to improve (see FQL(L)).

Importantly, ReFORM achieves the highest fraction on normalized scores close to 1, indicating that ReFORM does not limit the improvement of the learned policy as discussed in Section 4. The use of a support constraint allows the learned policy to apply any action with the support, including ones

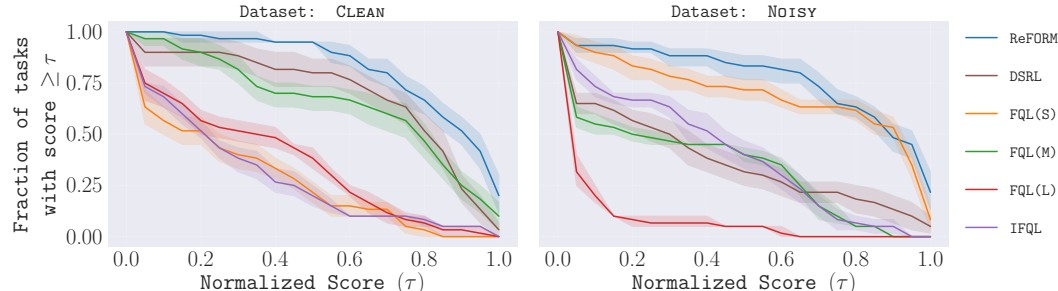

Figure 2: **Performance profile over CLEAN and NOISY datasets.** For a given normalized score $\tau$ (x-axis), the performance profile shows the probability that a given method achieves a score $\geq \tau$ (see Agarwal et al. (2021) for details). On the CLEAN dataset, ReFORM achieves greater scores with higher probabilities than all other baselines. The same is true on the NOISY dataset except for a small set of normalized scores around 0.9 where ReFORM and FQL(S) have similar probabilities within the statistical margins.

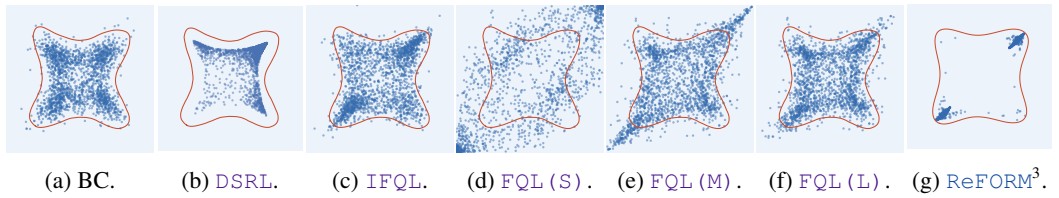

   (a) BC.     (b) DSRL.   (c) IFQL.   (d) FQL(S).  (e) FQL(M).  (f) FQL(L).  (g) ReFORM[3].

Figure 3: **Learned policy distributions with the toy example.** The $Q$-value reaches the maximum at the lower left and upper right corners (See the $Q$-value plot in Figure 1). The red boundaries denote the estimated $\text{supp}(\pi_{\text{BC}})$[4].

that have low density under the behavior policy $\pi_\beta$ Therefore, the learned policy does not suffer from a performance upper bound related to the behavior policy.

**(Q2): ReFORM maximizes the performance while avoiding OOD.** We design a toy example to better visualize and compare the learned policies. The toy example has a 2-dimensional action space with a $Q$-value that grows when approaching the lower left and the upper right corners (see the $Q$-value plot in Figure 1). We plot the policy distributions of BC and all algorithms in Figure 3. ReFORM maximizes performance by reaching both corners while staying within the support of the BC policy. DSRL collapses to a single mode in the upper right corner and remains far from the boundaries of the support because the generated noise of DSRL is unimodal and squashed. We compare the generated noise in more detail in Appendix C.4, Figure 16. IFQL remains similar to the BC policy because importance sampling is less efficient for finding the maximum. FQL faces OOD error due to its use of Wasserstein distance regularization (as discussed in Proposition 2).

### 5.3 ABLATION STUDIES

To study the functionality of each component of ReFORM, we conduct the following experiments in a toy environment and the `cube-single` environment with the NOISY dataset to answer Q3-Q5. All details can be found in Appendix C.3.3.

**(Q3): Having bounded support for the BC flow policy's source distribution is crucial.** We investigate the effect of satisfying support constraints (and hence the necessity of using a source distribution with bounded support) by using a Gaussian $\mathcal{N}(0, I^d)$ with unbounded support as the source distribution for both the BC flow policy and the flow noise generator following Wagenmaker et al. (2025) (ReFORM(U)). Figure 4 shows that ReFORM(U) suffers from severe OOD problems

---

[3]This plotted support slightly differs because $q_{\text{BC}} = \mathcal{U}(\mathcal{B}_l^d)$ for ReFORM, but $q_{\text{BC}} = \mathcal{N}(0, I^d)$ for others.
[4]The support estimation has some numerical errors, so a few samples of BC/IFQL can be outside.

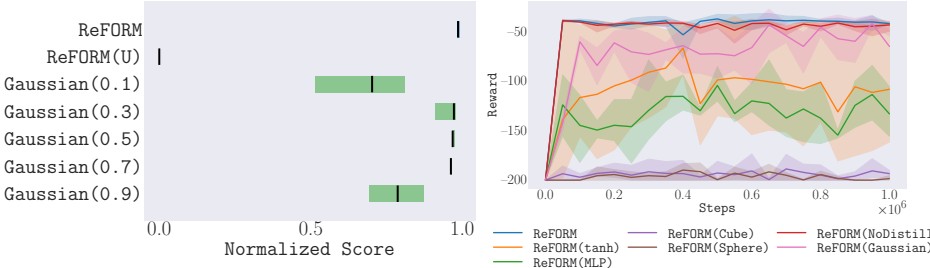

Figure 4: **Ablations.** Left: normalized scores of `ReFORM` and its variants with different source distributions. Right: training curves of `ReFORM` and its variants by changing its components.

and does not learn anything. This confirms that the ability of `ReFORM` to satisfy support constraints using a source distribution with bounded support is crucial to good performance.

We next change the source distribution of the flow noise generator of `ReFORM(U)` back to $\mathcal{U}(\mathcal{B}_l^d)$ while keeping $q_{\mathrm{BC}} = \mathcal{N}(0, I^d)$ for the BC flow policy. We also add the reflection term back to the noise generator. We change $l$ so that $\mathcal{B}_l^d$ is the $\xi$-confidence level of $q_{\mathrm{BC}}$. We vary $\xi$ within $\{0.1, 0.3, 0.5, 0.7, 0.9\}$ (`Gaussian(`$\xi$`)`). These baselines are highly sensitive to the choice of $\xi$, whereas `ReFORM` both avoids this additional hand-tuned hyperparameter $\xi$ and achieves better performance than the best performing `Gaussian(`$\xi$`)` (Figure 4).

**(Q4): The reflected flow improves the quality of the generated noise.** We consider replacing the reflected flow with three different generative models that also generate noise within the hypersphere $\mathcal{B}_l^d$: a MLP noise generator (`ReFORM(MLP)`), a "squashed flow" that applies a $\tanh$ at the end (`ReFORM(tanh)`), and a squashed Gaussian (`ReFORM(Gaussian)`) similar to `DSRL`. All baselines perform worse than `ReFORM` (Figure 4): the MLP and the Gaussian fail to capture multimodal distributions, while $\tanh$ squashing suffers from gradient vanishing.

**(Q5): Our design of the reflection term works the best within our considered choices.** We consider two other options for the reflection term. First, `ReFORM(Cube)` replaces our hypersphere-shaped domain $\mathcal{B}_l^d$ with a hypercube-shaped domain, while still applying the reflection term as introduced in Xie et al. (2024). Second, `ReFORM(Sphere)` shares our hypersphere-shaped domain, but instead of compensating the outbound velocity, it reflects the outbound velocity back inbound once the sample hits $\partial \mathcal{B}_l^d$. Figure 4 shows that these two variants cannot perform similarly to `ReFORM`. We hypothesize that compensating for the outbound velocities makes the training process more stable than reflecting the outbound velocities. We leave finding theoretical explanations of this phenomenon to future work.

**(Q6): Removing the BC flow policy distillation slightly degrades the performance of `ReFORM`.** We compare `ReFORM` with its variant `ReFORM(NoDistill)` by removing the distillation of the BC flow policy. Figure 4 shows that `ReFORM(NoDistill)`'s performance decreases slightly compared with `ReFORM`. This suggests that a longer backpropagation chain can be harmful, which matches the observation in Park et al. (2025b).

## 6    CONCLUSION

We propose `ReFORM` for realizing the support constraint with flow policies in offline RL. `ReFORM` simultaneously learns a BC flow policy that transforms a bounded uniform distribution in a hypersphere to the complex action distribution that matches the behavior policy, and a flow noise generator that transforms a bounded uniform distribution to a complex noise distribution being fed into the BC policy. With reflected flow on the noise generator, the noise generator is capable of generating complex multimodal noise while staying within the domain of the prior distribution of the BC policy. Therefore, `ReFORM` avoids the OOD issues by construction, putting no further constraints limiting the performance of the learned policy, and learns a complex multimodal policy. Our extensive experiments on 40 challenging tasks with the OGBench offline RL benchmark suggest that `ReFORM` achieves the best performance with only a single set of hyperparameters, eliminating the

costly fine-tuning process of most offline RL methods. The reflected flow noise generator can also be potentially combined with other generative-model-based policies, including diffusion policies.

**Limitations.** We identify several promising avenues for future work. Although our distillation step avoids BPTT through the BC flow, training the noise generator still relies on BPTT, which can be computationally intensive for deep models. This process can be potentially improved with shortcut models (Espinosa-Dice et al., 2025), or by applying a pre-trained BC model and latent space RL (Wagenmaker et al., 2025). Furthermore, our method ensures that the policy $\pi_\theta$ remains within the support of the BC policy, meaning that it inherits any potential OOD errors made by the BC model itself. Integrating behavior cloning methods with stricter support constraints, diagnosing when the BC model generates OOD errors, or applying a pre-trained BC model could mitigate this dependence. Moreover, the design of the reflection term is a nascent area, and exploring more adaptive or even learned reflection terms presents an exciting direction for developing more powerful policy improvement methods. In addition, ReFORM applies the simplest value function learning method and actor-critic structure similar to Park et al. (2025b), which can be potentially improved by other methods (Mao et al., 2023b; Garg et al., 2023; Liu et al., 2024; Agrawalla et al., 2025). Finally, ReFORM learns slower than algorithms imposing statistical distance regularization when the dataset contains expert policies due to the lack of any explicit regularization to keep the learned policy close to the expert policy.

## REPRODUCIBILITY STATEMENT

For better reproducibility, we provide all the proofs of theoretical results in Appendix A, and implementation details, including all hyperparameters in each environment of all algorithms in Appendix C.3. The benchmark we use is open-source and published in Park et al. (2025a). We also included the source code of our algorithm in the supplementary materials.

## ACKNOWLEDGEMENTS

This material is based upon work supported by the Under Secretary of Defense for Research and Engineering under Air Force Contract No. FA8702-15-D-0001 or FA8702-25-D-B002. In addition, Zhang, So, and Fan are supported by the National Science Foundation (NSF) CAREER Award #CCF-2238030. This work was also supported in part by a grant from Amazon. Any opinions, findings, conclusions or recommendations expressed in this material are those of the author(s) and do not necessarily reflect the views of the Under Secretary of Defense for Research and Engineering.

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

# A   PROOFS

## A.1   PROOF OF PROPOSITION 1

*Proof.* We first prove the first statement. We prove this by contradiction. Suppose $\mathrm{supp}(\pi_\theta(\cdot|s)) \not\subseteq \mathrm{supp}(\pi_\beta(\cdot|s))$. Then, there exists a region $\mathcal{B} = \{a \in \mathcal{A} \mid \pi_\theta(a|s) > 0, \pi_\beta(a|s) = 0\}$ with a non-zero measure. By the definition of the KL divergence, we have

$$D_{\mathrm{KL}}(\pi_\theta(\cdot|s) \mid\mid \pi_\beta(\cdot|s)) = \int_{a \in \mathcal{B}} \pi_\theta(a|s) \log \frac{\pi_\theta(a|s)}{\pi_\beta(a|s)} da + \int_{a \in \mathcal{A}\backslash\mathcal{B}} \pi_\theta(a|s) \log \frac{\pi_\theta(a|s)}{\pi_\beta(a|s)} da,$$
(14)

where the first term is $\infty$ and the second term is finite. Therefore, we have $D_{\mathrm{KL}}(\pi_\theta(\cdot|s) \mid\mid \pi_\beta(\cdot|s)) = \infty$, which contradicts the condition that $D_{\mathrm{KL}}(\pi_\theta(\cdot|s) \mid\mid \pi_\beta(\cdot|s)) \leq \epsilon < \infty$.

We then prove the second statement. Consider $\pi_\theta(\cdot|s) = \mathcal{N}(\mu, 1)$ and $\pi_\beta(\cdot|s) = \mathcal{N}(0, 1)$. We have $\mathrm{supp}(\pi_\theta(\cdot|s)) \subseteq \mathrm{supp}(\pi_\beta(\cdot|s))$. The KL divergence between them is

$$D_{\mathrm{KL}}(\pi_\theta(\cdot|s) \mid\mid \pi_\beta(\cdot|s)) = \frac{\mu^2}{2}.$$
(15)

Therefore, for any $M > 0$, we can choose $\mu > \sqrt{2M}$ so that $D_{\mathrm{KL}}(\pi_\theta(\cdot|s) \mid\mid \pi_\beta(\cdot|s)) > M$.   $\square$

## A.2   PROOF OF PROPOSITION 2

*Proof.* For simplicity, consider a given state $s \in \mathcal{S}$. We define $p_\beta(\cdot) = \pi_\beta(\cdot|s)$ and $p_\theta(\cdot) = \pi_\theta(\cdot|s)$. We prove by construction. We consider the optimal transport problem. First, we define a source region within the support of $p_\beta$. Consider a small ball $\mathcal{B}_1 \in \mathrm{supp}(p_\beta)$ centered at $a_1$. The probability mass in the ball is $\delta = \int_{\mathcal{B}_1} p_\beta(a) da$. Second, we define a target region. Consider another small ball $\mathcal{B}_2 \not\subset \mathrm{supp}(p_\beta)$ centered at $a_2$ with the same radius as $\mathcal{B}_1$. Let the distance between the two balls be $d = \|a_1 - a_2\|$. We define the new probability $p_\theta$ such that

$$p_\theta(a) = \begin{cases} p_\beta(a), & \text{if } a \notin \mathcal{B}_1 \text{ and } a \notin \mathcal{B}_2, \\ 0, & \text{if } a \in \mathcal{B}_1, \\ p_\beta(a - a_2 + a_1), & \text{if } a \in \mathcal{B}_2, \end{cases}$$
(16)

Then, we have $\mathrm{supp}(p_\theta) \not\subseteq \mathrm{supp}(p_\beta)$. We make $d \leq \sqrt{\frac{\epsilon^2}{\delta}}$ by choosing the source region $\mathcal{B}_1$ close to the boundary of $\mathrm{supp}(p_\beta)$ and the target region $\mathcal{B}_2$ close to $\mathcal{B}_1$. Then, we have

$$D_{\mathrm{W}2}(p_\theta \mid\mid p_\beta)^2 \leq \int_{a \in \mathcal{B}_1} \|d\|^2 p_\beta(a) da = d^2 \int_{a \in \mathcal{B}_1} p_\beta(a) da = d^2 \delta \leq \epsilon^2.$$
(17)

Therefore, we have $D_{\mathrm{W}2}(p_\theta \mid\mid p_\beta) \leq \epsilon$.   $\square$

## A.3   PROOF OF THEOREM 1

*Proof.* Remember that the source distribution of the BC flow policy is $q_{\mathrm{BC}} = \mathcal{U}(\mathcal{B}_l^d)$. We prove the theorem by showing that $z_k \in \mathcal{U}(\mathcal{B}_l^d)$ for all $k \in \{0, 1, \ldots, N-1\}$, which implies that $z \in \mathcal{B}_l^d$, for all $z \sim \tilde{q}_{\mathrm{BC}}$. We prove this by induction.

First, we have $z_0 = w \in \mathcal{B}_d^l$ because $w \sim \mathcal{U}(\mathcal{B}_l^d)$. Next, we assume that $z_k \in \mathcal{U}(\mathcal{B}_l^d)$. Then, we have the following two cases:

**Case 1:** $\|\hat{z}_{k+1}\| \leq l$. Following Eq. (12), we have $z_{k+1} = \hat{z}_{k+1} \in \mathcal{B}_l^d$.

**Case 2:** $\|\hat{z}_{k+1}\| > l$. Following Eq. (12), we have

$$\begin{aligned} z_{k+1} &= \hat{z}_{k+1} - \langle v_{\theta_2}(k\Delta t, w; s)\Delta t, n_{k+1}\rangle n_{k+1} \\ &= (\|\hat{z}_{k+1}\| - \langle v_{\theta_2}(k\Delta t, w; s)\Delta t, n_{k+1}\rangle) \, n_{k+1}. \end{aligned}$$
(18)

In addition, we have

$$\langle v_{\theta_2}(k\Delta t, w; s)\Delta t, n_{k+1}\rangle = \langle \hat{z}_{k+1} - z_k, n_{k+1}\rangle = \|\hat{z}_{k+1}\| - \langle z_k, n_{k+1}\rangle.$$
(19)

Plugging this into the previous equation, we get,

$$z_{k+1} = \langle z_k, n_{k+1} \rangle \, n_{k+1}. \tag{20}$$

Hence, we get,

$$\|z_{k+1}\| = |\langle z_k, n_{k+1} \rangle| \leq \|z_k\| \leq l \tag{21}$$

Thus our reflection ensures $z_{k+1} \in \mathcal{B}_d^l, \forall k$. Therefore, we have $z = z_N \in \mathcal{B}_l^d$, for all $z \sim \tilde{q}_{\mathrm{BC}}$. As a result, $\mathrm{supp}(\tilde{q}_{\mathrm{BC}}) \subseteq \mathrm{supp}(q_{\mathrm{BC}})$. $\qquad\square$

### A.4 PROOF OF THEOREM 2

*Proof.* Let $\tilde{z} \sim \tilde{q}_{\mathrm{BC}}$ be a sample from $\tilde{q}_{\mathrm{BC}}$. We have $\tilde{z} \in \mathrm{supp}(\tilde{q}_{\mathrm{BC}})$. Following Theorem 1, we have $\mathrm{supp}(\tilde{q}_{\mathrm{BC}}) \subseteq \mathrm{supp}(q_{\mathrm{BC}})$. Therefore, we have $\tilde{z} \in \mathrm{supp}(q_{\mathrm{BC}})$. Now consider the original target distribution $p_{\mathrm{BC}}$. Its support is the set of all points generated by applying the flow $\psi_{\theta_1}$ to all points in the support of $q_{\mathrm{BC}}$, i.e.,

$$\mathrm{supp}(p_{\mathrm{BC}}) = \{\psi_{\theta_1}(1, z; s) \mid z \in \mathrm{supp}(q_{\mathrm{BC}})\}. \tag{22}$$

Since we have $\tilde{z} \in \mathrm{supp}(q_{\mathrm{BC}})$, then by definition, we have $\psi_{\theta_1}(1, \tilde{z}; s) \in \mathrm{supp}(p_{\mathrm{BC}})$. This is true for all $\tilde{z} \sim \tilde{q}_{\mathrm{BC}}$. Therefore, by the definition of the support of $\tilde{p}_{\mathrm{BC}}$, i.e.,

$$\mathrm{supp}(\tilde{p}_{\mathrm{BC}}) = \{\psi_{\theta_1}(1, \tilde{z}; s) \mid \tilde{z} \in \mathrm{supp}(\tilde{q}_{\mathrm{BC}})\}, \tag{23}$$

we have $\mathrm{supp}(\tilde{p}_{\mathrm{BC}}) \subseteq \mathrm{supp}(p_{\mathrm{BC}})$. $\qquad\square$

## B ALGORITHM DETAILS

We provide the step-by-step explanation of ReFORM in Algorithm 1, where $\mathrm{RF}(v, s, w, N)$ means solving the reflected ODE (9) following the projected Euler step (12) with the velocity field $v$, state $s$, sample from the source distribution $w$, and number of Euler steps $N$.

## C EXPERIMENTS

### C.1 COMPUTATION RESOURCES

The experiments are run on a 13th Gen Intel(R) Core(TM) i7-13700KF CPU with 64GB RAM and an NVIDIA GeForce RTX 4090 GPU. The training time is around $80$ minutes for $10^6$ steps for ReFORM.

### C.2 ENVIRONMENTS

We conduct experiments on the recently published OGBench benchmark (Park et al., 2025a). We use $4$ environments (1 locomotion environment and 3 manipulation environments), $5$ tasks in each environment, with $2$ different datasets, for a total $40$ tasks. Since OGBench was originally designed for offline goal-conditioned RL, we use the single-task variants ("-singletask") for OGBench tasks to benchmark standard reward-maximizing offline RL. The reward functions in OGBench are semi-sparse. For the locomotion task, the reward functions are always $-1$ for not reaching the goal and $0$ for reaching the goal. Manipulation tasks usually contain several subtasks, and the rewards are bounded by $-n_{\mathrm{task}}$ and $0$, where $n_{\mathrm{task}}$ is the number of subtasks. All episodes end when the agent achieves the goal.

In our experiments, we consider the following tasks with the CLEAN dataset, where the demonstrations are randomly generated by an expert policy:

- `antmaze-large-navigate-singletask-task{1,2,3,4,5}-v0`
- `cube-single-play-singletask-task{1,2,3,4,5}-v0`
- `cube-double-play-singletask-task{1,2,3,4,5}-v0`
- `scene-play-singletask-task{1,2,3,4,5}-v0`

---

**Algorithm 1** ReFORM Algorithm

---

1: **Input:** Offline dataset $\mathcal{D}$; total Euler number of steps $N$, radius $l$
2: **Networks:** Critic $Q_\phi(s, a)$; BC flow field $v_{\theta_1}(t, z; s)$; noise flow field $v_{\theta_2}(t, w; s)$; one-step BC flow policy $\mu_{\hat{\theta}_1}(z; s)$.
3: **while** not converged **do**
4:  Sample batch $\{(s, a, r, s')\} \sim \mathcal{D}$
5:
6:  ▷ **Critic update**
7:  $w \sim \mathcal{U}(\mathcal{B}_l^d)$
8:  $z \leftarrow \mathrm{RF}(v_{\theta_2}, s', w, N)$
9:  $a' \leftarrow \mu_{\hat{\theta}_1}(z; s')$
10:  Update $\phi$ to minimize $\mathbb{E}\left[(r + \gamma Q_{\hat{\phi}}(s', a') - Q_\phi(s, a))^2)\right]$
11:
12:  ▷ **Train vector field $v_{\theta_1}$ in the BC flow policy $\mu_{\theta_1}$**
13:  $z \sim \mathcal{U}(\mathcal{B}_l^d)$
14:  $x_1 \leftarrow a$
15:  $t \sim \mathcal{U}[0, 1]$
16:  $x_t \leftarrow (1 - t)\, z + t\, x_1$
17:  Update $\theta_1$ to minimize $\mathbb{E}\left[\|v_{\theta_1}(t, x_t; s) - (x_1 - z)\|^2\right]$
18:
19:  ▷ **Train one-step policy $\mu_{\hat{\theta}_1}$**
20:  $z \sim \mathcal{U}(\mathcal{B}_l^d)$
21:  $a^{\mu_1} \leftarrow \mu_{\hat{\theta}_1}(z; s)$
22:  Update $\hat{\theta}_1$ to minimize $\mathbb{E}\left[\|a^{\mu_1} - \mu_{\theta_1}(z; s)\|^2\right]$
23:
24:  ▷ **Train vector field $v_{\theta_2}$ in the flow noise generator $\mu_{\theta_2}$**
25:  $w \sim \mathcal{U}(\mathcal{B}_l^d)$
26:  $z \leftarrow \mathrm{RF}(v_{\theta_2}, s, w, N)$
27:  $a^{\mu_2} \leftarrow \mu_{\theta_1}(z; s)$
28:  Update $\theta_2$ to minimize $\mathbb{E}\left[-Q_\phi(s, a^{\mu_2})\right]$

---

We also consider the NOISY dataset, where the demonstrations are randomly generated by a highly suboptimal and noisy policy:

- `antmaze-large-explore-singletask-task{1,2,3,4,5}-v0`

- `cube-single-noisy-singletask-task{1,2,3,4,5}-v0`

- `cube-double-noisy-singletask-task{1,2,3,4,5}-v0`

- `scene-noisy-singletask-task{1,2,3,4,5}-v0`

More details about the environment and videos of the demonstrations can be found in the OGBench paper (Park et al., 2025a).

## C.3 IMPLEMENTATION DETAILS AND HYPERPARAMETERS

### C.3.1 DETAILS OF REFORM

**Flow policies.** We parameterize the velocity fields of the BC flow policy $v_{\theta_1}$ and the flow noise generator $v_{\theta_2}$ with MLPs. We use the Euler method to solve ODE (4) for the BC flow policy, and the projected Euler step (12) to solve the reflected ODE (9) for the flow noise generator. 10 Euler steps are used for both Euler integration for all environments.

**$Q$-functions.** Following the standard implementation of $Q$-functions in RL, we train two $Q$ functions to improve stability. Two aggregation methods are used to aggregate the two $Q$-values for different environments following Park et al. (2025b). For most environments, we take the mean of the two $Q$-values for aggregation (Ball et al., 2023; Nauman et al., 2024), except for the

Table 1: Training steps for all algorithms for each task.

| Task | Dataset | Training step |
|------|---------|---------------|
| antmaze-large-navigate-singletask-task{1,2,3,4,5}-v0 | CLEAN | $1 \times 10^7$ |
| antmaze-large-explore-singletask-task{1,2,3,4,5}-v0 | NOISY | $8 \times 10^6$ |
| cube-single-play-singletask-task{1,2,3,4,5}-v0 | CLEAN | $2 \times 10^6$ |
| cube-single-noisy-singletask-task{1,2,3,4,5}-v0 | NOISY | $3 \times 10^6$ |
| cube-double-play-singletask-task{1,2,3,4,5}-v0 | CLEAN | $2 \times 10^6$ |
| cube-double-noisy-singletask-task{1,2,3,4,5}-v0 | NOISY | $1 \times 10^6$ |
| scene-play-singletask-task1-v0 | CLEAN | $2 \times 10^6$ |
| scene-play-singletask-task{2,3,4,5}-v0 | CLEAN | $3 \times 10^6$ |
| scene-noisy-singletask-task{1,2}-v0 | NOISY | $1 \times 10^6$ |
| scene-noisy-singletask-task{3,4,5}-v0 | NOISY | $2 \times 10^6$ |

Table 2: Common hyperparameters for all algorithms.

| Hyperparameter | Value |
|----------------|-------|
| Learning rate | 0.0003 |
| Optimizer | Adam (Kingma & Ba, 2015) |
| Maximum gradient norm | 10 |
| Target network smoothing coefficient | 0.005 |
| Discount factor $\gamma$ | 0.995 |
| MLP dimensions | $[512, 512, 512, 512]$ |
| Nonlinearity | GELU (Hendrycks & Gimpel, 2016) |
| Flow steps | 10 |
| Flow time sampling distribution | $\mathcal{U}[0, 1]$ |
| Minibatch size | 256 |
| Clipped double $Q$-learning | False (default), True (`antmaze-large`) |

`antmaze-large` environment, where we take the minimum of the two $Q$-values (Van Hasselt et al., 2016; Fujimoto et al., 2018).

**Selection of the radius of the hypersphere $\mathcal{B}_l^d$.** As the action space for physical systems is always compact, we select the hypersphere $\mathcal{B}_l^d$ to be the smallest hypersphere that contains the action space, i.e., $l = \min_{l'}\{l' \in \mathbb{R}^d \mid \mathcal{A} \subseteq \mathcal{B}_{l'}^d\}$. Note that, as the action space $\mathcal{A}$ is known and is usually a hyperbox, in most cases, we can compute the solution easily, or, otherwise, use an overapproximation of $\mathcal{B}_l^d$. Therefore, this choice does not impose any limitation on our approach. We also present experimental results of the sensitivity of ReFORM w.r.t. $l$ in Appendix C.4.

**Neural Network architectures.** For all neural networks in our experiments, we use MLPs with 4 hidden layers and 512 neurons on each layer. We apply layer normalization (Ba et al., 2016) to the $Q$-function networks to stabilize training.

**Training and evaluation.** The difficulty of tasks in OGBench can be very different. Therefore, we use different training steps for different tasks (Table 1). For each task, we train each algorithm with 3 different seeds and evaluate the model saved at the last epoch for 32 episodes.

### C.3.2 DETAILS OF BASELINES IN MAIN RESULTS

We choose the state-of-the-art offline RL methods with flow policies as our baselines, including FQL (Park et al., 2025b), IFQL (Hansen-Estruch et al., 2023b; Park et al., 2025b), and DSRL (Wagenmaker et al., 2025). We implement the baselines FQL and IFQL following the original implementation provided in Park et al. (2025b), and DSRL also following the original implementation provided in Wagenmaker et al. (2025).

Table 3: Environment-specific hyperparameters for `FQL` and `DSRL`.

| Environment | `FQL(S)` $\alpha$ | `FQL(M)` $\alpha$ | `FQL(L)` $\alpha$ | Noise bound for `DSRL` |
|---|---|---|---|---|
| `antmaze-large` | 1 | 10 | 100 | $[-1.25, 1.25]$ |
| `cube-single` | 30 | 300 | 3000 | $[-0.5, 0.5]$ |
| `cube-double` | 30 | 300 | 3000 | $[-1.5, 1.5]$ |
| `scene` | 30 | 300 | 3000 | $[-0.75, 0.75]$ |

Figure 5: Normalized scores with the CLEAN dataset.

### C.3.3 DETAILS OF BASELINES IN ABLATION STUDIES

**ReFORM(U).** ReFORM(U) modifies ReFORM by changing the source distribution of both the BC policy and the noise generator from $\mathcal{U}(\mathcal{B}_l^d)$ to $\mathcal{N}(0, I^d)$. In other words, we have $q_{\mathrm{NG}} = q_{\mathrm{BC}} = \mathcal{N}(0, I^d)$ for ReFORM(U).

**Gaussian($\xi$).** Gaussian($\xi$) modifies ReFORM by changing the source distribution of the BC policy from $\mathcal{U}(\mathcal{B}_l^d)$ to $\mathcal{N}(0, I^d)$, then choose $l$ so that $\mathcal{B}_l^d$ is the $\xi$-confidence level of $\mathcal{N}(0, I^d)$, i.e., $l = \sqrt{\mathrm{PPF}_{\chi_d^2}(\xi)}$, where $\mathrm{PPF}_{\chi_d^2}$ is the percent point function of a $d$-dimensional $\chi^2$ distribution.

**ReFORM(MLP).** ReFORM(MLP) modifies ReFORM by changing the reflected flow noise generator to an MLP noise generator $f(s) : \mathcal{S} \to \mathcal{B}_l^d$, which maps the state to a point within $\mathrm{supp}(q_{\mathrm{BC}})$.

**ReFORM($\tanh$).** ReFORM($\tanh$) modifies ReFORM by removing the reflection term in the reflection ODE (9), i.e., using (11) instead of (12) when integrating the noise generator flow. Then, after the Euler integration and getting $\hat{z}$ following (11), we use $\tanh$ to squash the norm of $z$ so that it stays within $\mathcal{B}_l^d$. In other words, $z = \frac{\hat{z}}{\|\hat{z}\|} \cdot \tanh(\|\hat{z}\|) \cdot l$.

**ReFORM(cube).** ReFORM(cube) modifies ReFORM by changing the domain of $q_{\mathrm{NG}}$ and $q_{\mathrm{BC}}$ to $[-1, 1]^d$. Then, the reflected ODE is solved by first using the Euler integration (11) to get $\hat{z}$, and then applying $z = 1 - |(\hat{z} + 1) \bmod 4 - 2|$ following Xie et al. (2024).

**ReFORM(sphere).** ReFORM(sphere) modifies ReFORM by changing the reflection term from compensating the outbound velocity to "bouncing back", like billiards.

**ReFORM(NoDistill).** ReFORM(NoDistill) removes the distillation part of ReFORM, i.e., the actor loss (10) is backpropagated through the BC flow policy instead of the one-step policy.

### C.3.4 HYPERPARAMETERS

The choice of hyperparameters largely follows Park et al. (2025b). We provide the common hyperparameters shared for all algorithms in Table 2, and the environment-specific hyperparameters for `FQL` and `DSRL` in Table 3. Note that all environment-specific hyperparameters for `FQL(M)` and `DSRL` are the same as provided in their original papers (with the CLEAN dataset), which are hand-tuned for each environment. As the baselines were not tested on the NOISY dataset in their original papers, we use the same hyperparameters for them in the same environment with the CLEAN dataset.

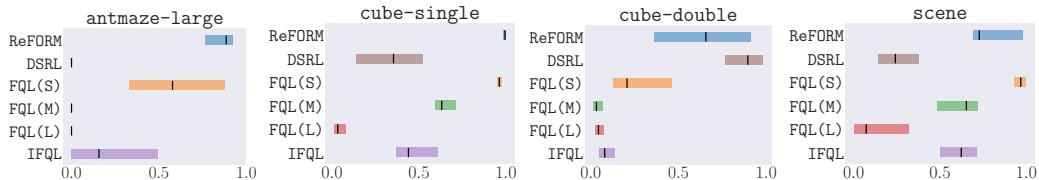

Figure 6: Normalized scores with the NOISY dataset.

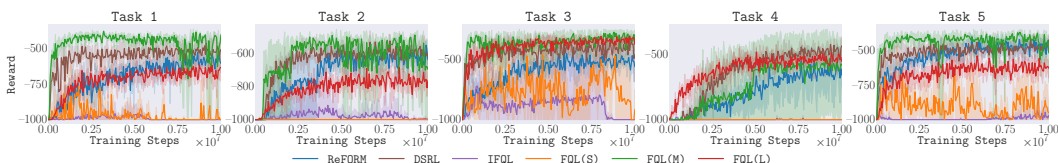

Figure 7: Training curves for `antmaze-large` environment with the CLEAN dataset.

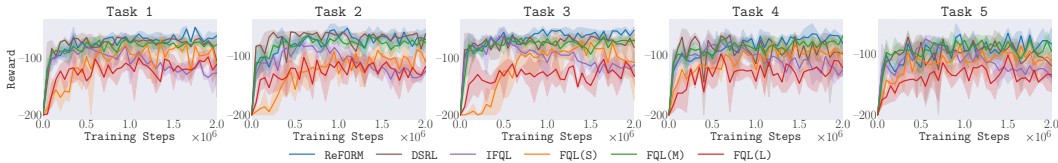

Figure 8: Training curves for `cube-single` environment with the CLEAN dataset.

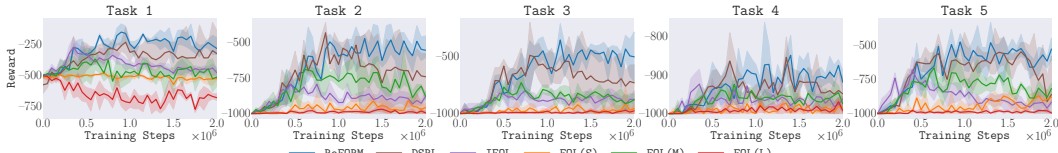

Figure 9: Training curves for `cube-double` environment with the CLEAN dataset.

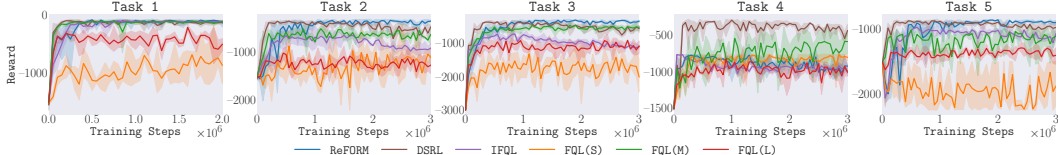

Figure 10: Training curves for `scene` environment with the CLEAN dataset.

## C.4 ADDITIONAL RESULTS

**Normalized scores for each environment and each dataset.** We present bar plots of the interquantile means (IQM) (see Agarwal et al. (2021) for details) of the normalized scores for each algorithm in each environment with the CLEAN dataset (Figure 5) and the NOISY dataset (Figure 6). We can observe that ReFORM consistently achieves the best or comparable results in all environments with both datasets, with a constant set of hyperparameters. DSRL and FQL(M) generally perform the second and third best in environments with the CLEAN dataset. However, their performance drops when the NOISY dataset is used.

**Full results.** We present the full per-task results of all 40 tasks in Table 4. The results are averaged over 3 seeds and 32 runs per seed. The results are bolded if the algorithm achieves at or above 95% of the best performance following Park et al. (2025a).

Table 4: **Full results.** We present full results (normalized score) on 40 OGBench tasks. The results are averaged over 3 seeds and 32 runs per seed. The results are bolded if the algorithm achieves at or above 95% of the best performance following Park et al. (2025a). To save space, the -singletask tags are omitted from task names.

| Task | Dataset | IFQL | FQL(L) | FQL(M) | FQL(S) | DSRL | ReFORM |
|------|---------|------|--------|--------|--------|------|--------|
| antmaze-large-navigate-task1-v0 | CLEAN | 0±0 | 51±2 | **96±3** | 1±1 | 85±9 | 65±9 |
| antmaze-large-navigate-task2-v0 | CLEAN | 0±0 | 56±4 | 62±44 | 0±0 | **83±9** | 72±5 |
| antmaze-large-navigate-task3-v0 | CLEAN | 0±0 | **91±6** | 90±7 | 8±11 | 77±9 | 61±2 |
| antmaze-large-navigate-task4-v0 | CLEAN | 0±0 | 74±5 | 67±45 | 0±0 | **80±6** | 68±12 |
| antmaze-large-navigate-task5-v0 | CLEAN | 2±2 | 61±4 | 76±26 | 6±4 | **86±9** | 90±3 |
| antmaze-large-explore-task1-v0 | NOISY | 40±30 | 0±0 | 0±0 | 84±7 | 0±0 | **91±6** |
| antmaze-large-explore-task2-v0 | NOISY | 0±0 | 0±0 | 0±0 | 41±13 | 0±0 | **91±6** |
| antmaze-large-explore-task3-v0 | NOISY | 69±28 | 0±0 | 0±0 | **92±8** | 0±0 | 87±2 |
| antmaze-large-explore-task4-v0 | NOISY | 36±15 | 0±0 | 0±0 | **56±37** | 0±0 | 5±8 |
| antmaze-large-explore-task5-v0 | NOISY | 0±0 | 0±0 | 0±0 | 16±22 | 0±0 | **88±14** |
| cube-single-play-task1-v0 | CLEAN | 40±13 | 47±5 | 86±2 | 57±12 | 60±43 | **97±3** |
| cube-single-play-task2-v0 | CLEAN | 7±3 | 20±17 | 73±10 | 57±5 | **86±3** | 85±11 |
| cube-single-play-task3-v0 | CLEAN | 14±5 | 18±1 | 77±6 | 44±35 | 68±21 | **99±1** |
| cube-single-play-task4-v0 | CLEAN | 30±6 | 19±17 | 73±9 | 77±3 | 59±19 | **89±11** |
| cube-single-play-task5-v0 | CLEAN | 43±12 | 25±19 | 85±14 | 73±3 | 61±28 | **97±4** |
| cube-single-noisy-task1-v0 | NOISY | 46±10 | 12±2 | 68±8 | 95±1 | 31±22 | **99±1** |
| cube-single-noisy-task2-v0 | NOISY | 53±15 | 2±2 | 71±3 | 97±1 | 45±10 | **100±0** |
| cube-single-noisy-task3-v0 | NOISY | 68±6 | 5±5 | 54±3 | **98±1** | 3±1 | 98±2 |
| cube-single-noisy-task4-v0 | NOISY | 40±4 | 2±1 | 63±5 | 94±1 | 31±5 | **100±1** |
| cube-single-noisy-task5-v0 | NOISY | 37±4 | 3±2 | 72±7 | **96±1** | 61±3 | 99±1 |
| cube-double-play-task1-v0 | CLEAN | 42±6 | 7±5 | 37±7 | 32±2 | 68±26 | **74±6** |
| cube-double-play-task2-v0 | CLEAN | 22±10 | 4±1 | 30±3 | 2±3 | 47±33 | **90±12** |
| cube-double-play-task3-v0 | CLEAN | 17±2 | 1±1 | 17±6 | 4±4 | 42±30 | **90±7** |
| cube-double-play-task4-v0 | CLEAN | 30±15 | 11±11 | 25±6 | 4±1 | 30±23 | **90±7** |
| cube-double-play-task5-v0 | CLEAN | 12±5 | 2±1 | 24±10 | 26±7 | 17±23 | **82±21** |
| cube-double-noisy-task1-v0 | NOISY | 62±5 | 6±4 | 12±14 | 68±14 | 86±9 | **94±6** |
| cube-double-noisy-task2-v0 | NOISY | 5±3 | 2±1 | 2±1 | 37±18 | **76±25** | 56±20 |
| cube-double-noisy-task3-v0 | NOISY | 5±3 | 3±2 | 7±6 | 15±4 | **75±30** | 52±22 |
| cube-double-noisy-task4-v0 | NOISY | 10±3 | 6±2 | 7±3 | 20±12 | **72±31** | 33±47 |
| cube-double-noisy-task5-v0 | NOISY | 8±6 | 6±4 | 2±0 | 9±6 | **90±10** | 67±23 |
| scene-play-task1-v0 | CLEAN | **99±1** | 69±6 | **98±2** | 23±17 | **94±2** | **95±2** |
| scene-play-task2-v0 | CLEAN | 41±7 | 8±6 | 73±3 | 27±12 | 87±1 | **99±1** |
| scene-play-task3-v0 | CLEAN | 56±2 | 52±4 | 89±1 | 20±15 | 85±2 | **99±1** |
| scene-play-task4-v0 | CLEAN | 24±1 | 15±11 | 65±26 | 46±2 | **97±2** | 25±10 |
| scene-play-task5-v0 | CLEAN | 80±3 | 65±5 | 76±6 | 22±23 | 92±3 | **99±1** |
| scene-noisy-task1-v0 | NOISY | 87±9 | 27±11 | 87±1 | **100±0** | 39±43 | **99±1** |
| scene-noisy-task2-v0 | NOISY | 40±7 | 1±1 | 23±5 | **95±4** | 15±4 | 61±14 |
| scene-noisy-task3-v0 | NOISY | 66±3 | 3±3 | 66±4 | **96±3** | 45±15 | 69±3 |
| scene-noisy-task4-v0 | NOISY | 57±4 | 5±7 | 53±6 | **88±8** | 25±10 | 71±0 |
| scene-noisy-task5-v0 | NOISY | 57±21 | 59±1 | 70±2 | **96±1** | 24±17 | **98±2** |

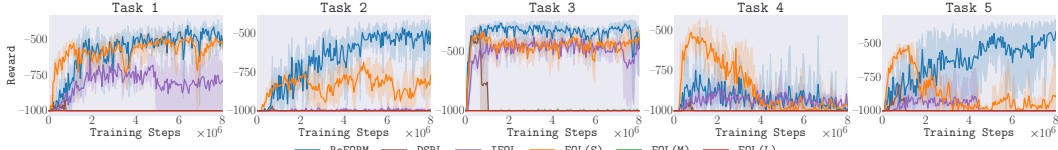

Figure 11: Training curves for antmaze-large environment with the NOISY dataset.

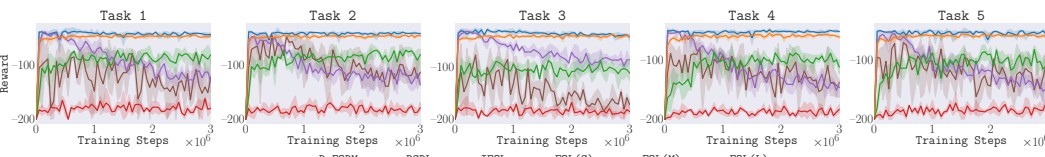

Figure 12: Training curves for cube-single environment with the NOISY dataset.

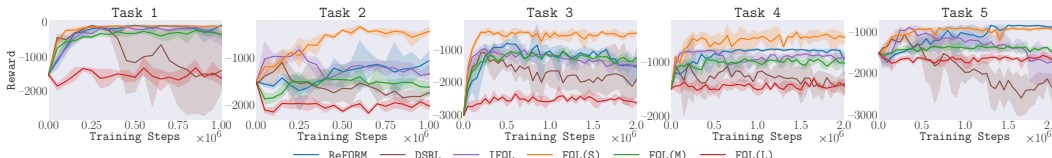

Figure 13: Training curves for `cube-double` environment with the NOISY dataset.

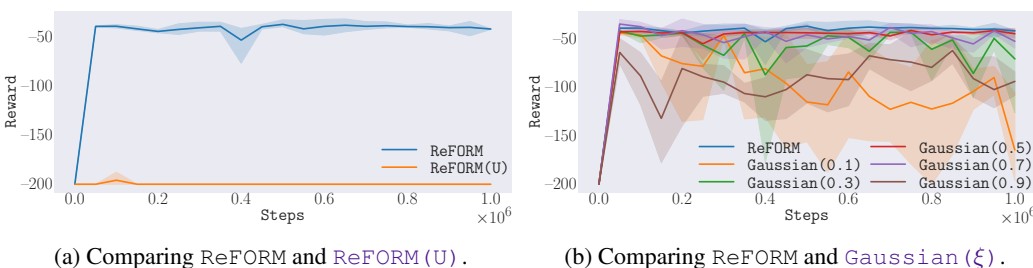

Figure 14: Training curves for `scene` environment with the NOISY dataset.

(a) Comparing `ReFORM` and `ReFORM(U)`.  (b) Comparing `ReFORM` and `Gaussian(ξ)`.

Figure 15: **More training curves for ablation studies.** `ReFORM(U)` changes both $q_{NG}$ and $q_{BC}$ from $\mathcal{U}(\mathcal{B}_l^d)$ to the standard Gaussian distribution and removes the reflection term. `Gaussian(ξ)` keeps $q_{NG} = \mathcal{U}(\mathcal{B}_l^d)$ but changes $q_{BC}$ to the standard Gaussian distribution. Then, the radius of the hypersphere $\mathcal{B}_l^d$ is chosen such that the standard Gaussian distribution has probability mass $\xi$ in $\mathcal{B}_l^d$.

**Training curves.** We present the training curves for all tasks in all environments with both the CLEAN dataset (Figure 7-Figure 10) and the NOISY dataset (Figure 11-Figure 14). In addition, we present training curves corresponding to Figure 4 (left) in the main pages in Figure 15.

**D4RL results.** We further conduct experiments in D4RL (Fu et al., 2020) antmaze and adoit environments to test the performance of `ReFORM` across different benchmarks. Although the benchmark is different, we *maintain* the hyperparameters of `ReFORM` as those in OGBench. The results (D4RL normalized return) are shown in Table 5, and `ReFORM` still consistently achieves the best or comparable results. `DSRL` is omitted in these results because its paper (Wagenmaker et al., 2025) does not report the best hyperparameters of `DSRL` in these environments.

**Visual manipulation results.** We also conduct experiments in OGBench (Park et al., 2025a) visual manipulation environments to test the performance of `ReFORM` with higher-dimensional image-based inputs. Similarly, we *maintain* the hyperparameters of `ReFORM`. The results (return) are shown in Table 6, and `ReFORM` performs the best. `DSRL` is omitted because its best hyperparameters in these environments are not reported in Wagenmaker et al. (2025).

**Visualization of the generated noise in the toy example.** For the toy example presented in Section 5.3, it is possible to visualize the generated noises directly for `ReFORM` and `DSRL`. We visualize the noises in Figure 16. We observe that the generated noise with reflected flow (`ReFORM`) is more concentrated while retaining two modes, while with a Gaussian distribution squashed by $\tanh$ (`DSRL`), the generated noise is unimodal and spreads out a lot.

**Ablations on the radius of the hypersphere $\mathcal{B}_l^d$.** One hyperparameter introduced in `ReFORM` is the radius $l$ of the hypersphere $\mathcal{B}_l^d$. As discussed in Appendix C.3, we select the smallest $l$ such

Table 5: **D4RL results.** We present the following results on environments in the D4RL Fu et al. (2020) benchmark. The results are averaged over 3 seeds and 32 runs per seed. The results are bolded if the algorithm achieves at or above 95% of the best performance following Park et al. (2025a).

| Environment | IFQL | FQL(L) | FQL(M) | FQL(S) | ReFORM |
|---|---|---|---|---|---|
| antmaze-umaze-v2 | $91_{\pm 7}$ | $85_{\pm 4}$ | $\mathbf{99}_{\pm 1}$ | $88_{\pm 13}$ | $\mathbf{97}_{\pm 0}$ |
| antmaze-umaze-diverse-v2 | $55_{\pm 28}$ | $57_{\pm 10}$ | $\mathbf{88}_{\pm 5}$ | $61_{\pm 26}$ | $\mathbf{83}_{\pm 3}$ |
| antmaze-medium-play-v2 | $3_{\pm 4}$ | $14_{\pm 6}$ | $\mathbf{92}_{\pm 1}$ | $52_{\pm 15}$ | $85_{\pm 4}$ |
| antmaze-medium-diverse-v2 | $24_{\pm 34}$ | $9_{\pm 4}$ | $\mathbf{81}_{\pm 13}$ | $24_{\pm 30}$ | $80_{\pm 4}$ |
| antmaze-large-play-v2 | $17_{\pm 21}$ | $43_{\pm 10}$ | $61_{\pm 21}$ | $3_{\pm 4}$ | $\mathbf{71}_{\pm 4}$ |
| antmaze-large-diverse-v2 | $28_{\pm 27}$ | $55_{\pm 4}$ | $\mathbf{85}_{\pm 8}$ | $8_{\pm 12}$ | $69_{\pm 9}$ |
| pen-human-v1 | $\mathbf{65}_{\pm 1}$ | $48_{\pm 0}$ | $59_{\pm 4}$ | $31_{\pm 4}$ | $\mathbf{64}_{\pm 7}$ |
| pen-cloned-v1 | $\mathbf{81}_{\pm 8}$ | $61_{\pm 7}$ | $66_{\pm 5}$ | $57_{\pm 6}$ | $70_{\pm 6}$ |
| pen-expert-v1 | $120_{\pm 3}$ | $105_{\pm 7}$ | $\mathbf{128}_{\pm 1}$ | $107_{\pm 10}$ | $\mathbf{129}_{\pm 7}$ |
| door-human-v1 | $3_{\pm 1}$ | $2_{\pm 1}$ | $0_{\pm 0}$ | $0_{\pm 0}$ | $\mathbf{4}_{\pm 1}$ |
| door-cloned-v1 | $-0_{\pm 0}$ | $0_{\pm 0}$ | $\mathbf{3}_{\pm 2}$ | $0_{\pm 0}$ | $1_{\pm 1}$ |
| door-expert-v1 | $89_{\pm 5}$ | $\mathbf{104}_{\pm 1}$ | $\mathbf{105}_{\pm 0}$ | $\mathbf{102}_{\pm 0}$ | $\mathbf{104}_{\pm 4}$ |

Table 6: **Visual manipulation results.** We present the following results on visual manipulation environments in OGBench Park et al. (2025a). The results are averaged over 3 seeds and 32 runs per seed. The results are bolded if the algorithm achieves at or above 95% of the best performance following Park et al. (2025a). To save space, the `-singletask` tags are omitted from task names.

| Task | Dataset | IFQL | FQL(L) | FQL(M) | FQL(S) | ReFORM |
|---|---|---|---|---|---|---|
| visual-cube-single-play-task1-v0 | CLEAN | $-117_{\pm 7}$ | $-150_{\pm 16}$ | $\mathbf{-110}_{\pm 9}$ | $-138_{\pm 19}$ | $\mathbf{-108}_{\pm 12}$ |
| visual-cube-single-noisy-task1-v0 | NOISY | $-95_{\pm 2}$ | $-176_{\pm 10}$ | $-103_{\pm 2}$ | $-57_{\pm 3}$ | $\mathbf{-52}_{\pm 7}$ |

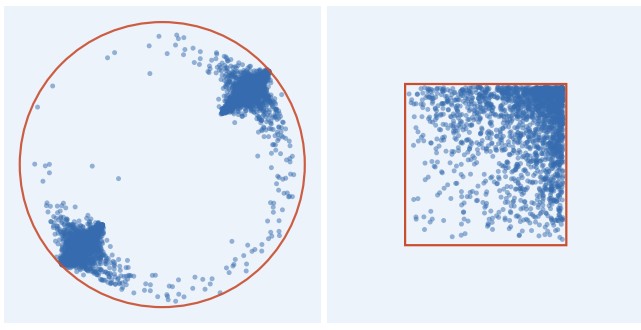

(a) Generated noise of ReFORM.  (b) Generated noise of DSRL.

Figure 16: Visualization of the generated noises in the 2-dimensional toy example.

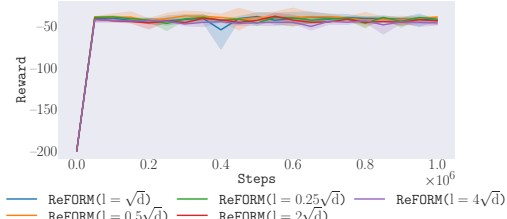

Figure 17: Sensitivity analysis of ReFORM w.r.t. $l$.

Table 7: Approximate training time of ReFORM and the baselines.

| Algorithm | ReFORM | FQL | IFQL | DSRL |
|---|---|---|---|---|
| Training time (minutes, $10^6$ steps) | 80 | 40 | 35 | 55 |

that $\mathcal{A} \subseteq \mathcal{B}_l^d$ in our implementation. In OGBench environments, the action space is $[-1, 1]^d$, so we choose $l = \sqrt{d}$. To study the sensitivity of ReFORM w.r.t. $l$, we conduct experiments in the cube-single environment with the NOISY dataset, and vary $l$ in $\{0.25\sqrt{d}, 0.5\sqrt{d}, \sqrt{d}, 2\sqrt{d}, 4\sqrt{d}\}$. The results are shown in Figure 17, which shows no significant difference among these choices. Therefore, ReFORM is robust w.r.t. the choice of $l$, and empirically $l$ can be chosen as any number close to the scale of the action space.

**Training time.** We report the training time of ReFORM and all baselines in Table 7. The table shows that ReFORM indeed doubles the training time compared to FQL due to the 2-stage flow. However, as shown in our experiments, FQL is sensitive to hyperparameters, and searching for optimal hyperparameters requires significantly more runs. On the contrary, ReFORM can be used without any hyperparameter searching.

## C.5 CODE

We provide code for ReFORM in our supplementary materials.

## D THE USE OF LARGE LANGUAGE MODELS

This paper uses Large Language Models to correct spelling and grammar issues.

