# OpenReview forum: "ReFORM: Reflected Flows for On-support Offline RL via Noise Manipulation"
_ICLR.cc/2026/Conference — ICLR 2026 Poster_

### Official Review · Reviewer_NkRQ · 2025-10-31

**Soundness:** 3
**Presentation:** 3
**Contribution:** 3
**Rating:** 6
**Confidence:** 2

**Summary:**

The paper proposes ReFORM, an offline RL method that builds a flow-policy constrained to the support of the behavior policy by construction. It first learns a behavior-cloned flow policy whose source distribution is a bounded hypersphere, then manipulates the latent noise via a reflected flow so that the pushforward actions remain on-support while still allowing expressive, multimodal policies. The authors prove that the reflected flow preserves support and thus avoids OOD actions without statistical-distance regularization, and they report strong results on 40 OGBench tasks using a single hyperparameter setting, with performance profiles outperforming flow-based baselines.

**Strengths:**

The method guarantees that the manipulated latent distribution and the resulting action distribution stay within the BC support (Theorems 1–2), directly addressing OOD actions without tuning a divergence weight.

Adapting reflected ODEs to latent noise yields an expressive, multimodal policy class while rigorously maintaining the bounded domain.

By dropping statistical-distance regularization, ReFORM uses constant hyperparameters across tasks, yet performs competitively or better on performance profiles.

Results across 40 tasks (clean/noisy datasets) show consistent gains against FQL/IFQL/DSRL, with informative ablations (source distribution, reflection design, distillation).

**Weaknesses:**

The work does not report results on widely used D4RL benchmarks (e.g., Locomotion/AntMaze v2), which would strengthen comparability with prior offline RL papers beyond OGBench. (Authors evaluate OGBench tasks instead.)

Dependence on BC support quality. Because actions are restricted to the BC support, any mis-estimated or overly conservative BC support could cap achievable performance; the paper could analyze this failure mode more explicitly (e.g., diagnostics when the BC model under-covers optimal modes)

**Questions:**

See weaknesses.

---

> ### Author Response · Authors · 2025-11-21
> **Author Reply (1/2)**
>
> We thank the reviewer for acknowledging that ReFORM avoids OOD actions without tuning a divergence weight, acknowledging the reflected flow's ability for generating expressive, multimodal noise, and recognizing our strong empirical results and thorough ablations.
>
> We hope our responses below and our revised manuscript address the concerns raised by the reviewer.
>
> ## Summary
>
> In brief, we have:
>
> 1. Provided additional experimental results with the D4RL antmaze, D4RL adroit, and OGBench visual manipulation benchmarks.
> 2. Clarified the dependence on the BC policy as a limitation of our work and all other offline RL works, and discussed possible solutions.
>
> ## Detailed Reply
>
> > **W1:** The work does not report results on widely used D4RL benchmarks (e.g., Locomotion/AntMaze v2), which would strengthen comparability with prior offline RL papers beyond OGBench. (Authors evaluate OGBench tasks instead.)
>
> Thanks for the suggestion! We agree that more experiments across different benchmarks can strengthen our paper, and **conducted additional experiments** on the D4RL benchmark and OGBench visual manipulation (with image-based inputs) tasks. The results are shown below:
>
> **D4RL antmaze (normalized return):**
>
> | Environment               | IFQL              | FQL (L)   | FQL (M)            | FQL (S)    | ReFORM             |
> |---------------------------|-------------------|-----------|--------------------|------------|--------------------|
> | antmaze-umaze-v2          | $91\pm7$          | $85\pm4$  | $\mathbf{99}\pm1$  | $88\pm13$  | $\mathbf{97}\pm0$  |
> | antmaze-umaze-diverse-v2  | $55\pm28$         | $57\pm10$ | $\mathbf{88}\pm5$  | $61\pm26$  | $\mathbf{83}\pm3$  |
> | antmaze-medium-play-v2    | $3\pm4$           | $14\pm6$  | $\mathbf{92}\pm1$  | $52\pm15$  | $85\pm4$           |
> | antmaze-medium-diverse-v2 | $24\pm34$         | $9\pm4$   | $\mathbf{81}\pm13$ | $24\pm30$  | $\mathbf{80}\pm4$  |
> | antmaze-large-play-v2     | $17\pm21$         | $43\pm10$ | $61\pm21$          | $3\pm4$    | $\mathbf{71}\pm4$  |
> | antmaze-large-diverse-v2  | $28\pm27$         | $55\pm4$  | $\mathbf{85}\pm8$  | $8\pm12$   | $69\pm9$           |
>
> **D4RL adroit (normalized return):**
>
> | Environment               | IFQL              | FQL (L)   | FQL (M)            | FQL (S)    | ReFORM             |
> |---------------------------|-------------------|-----------|--------------------|------------|--------------------|
> | pen-human-v1              | $\mathbf{65}\pm1$ | $48\pm0$  | $59\pm4$           | $31\pm4$   | $\mathbf{64}\pm7$  |
> | pen-cloned-v1             | $\mathbf{81}\pm8$ | $61\pm7$  | $66\pm5$           | $57\pm6$   | $70\pm6$           |
> | pen-expert-v1             | $120\pm3$         | $105\pm7$ | $\mathbf{128}\pm1$ | $107\pm10$ | $\mathbf{129}\pm7$ |
> | door-human-v1             | $3\pm1$           | $2\pm1$   | $0\pm0$            | $0\pm0$    | $\mathbf{4}\pm1$   |
> | door-cloned-v1            | $-0\pm0$          | $0\pm0$   | $\mathbf{3}\pm2$   | $0\pm0$    | $1\pm1$            |
> | door-expert-v1            | $89\pm5$          | $\mathbf{104}\pm1$ | $\mathbf{105}\pm0$ | $\mathbf{102}\pm0$ | $\mathbf{104}\pm4$ |
>
> **OGBench visual manipulation results (return):**
>
> |Task|Dataset|IFQL|FQL(L)|FQL(M)|FQL(S)|ReFORM|
> |--- |---    |--- |---   |---   |---   |---   |
> |visual-cube-single-play-singletask-task1-v0|CLEAN| $-117\pm7$ | $-150\pm16$ | $\mathbf{-110}\pm9$ | $-138\pm19$ | $\mathbf{-108}\pm12$ |
> |visual-cube-single-noisy-singletask-task1-v0|NOISY| $-95\pm2$  | $-176\pm10$ | $-103\pm2$          | $-57\pm3$   | $\mathbf{-52}\pm7$   |
>
> The results suggest that ReFORM, with a **constant** set of hyperparameters, can solve tasks across different benchmarks and even with high-dimensional image-based inputs.
> We have added all the new results and discussions in Appendix C.4 in our revision.

---

> > ### Author Response · Authors · 2025-11-21
> > **Author Reply (2/2)**
> >
> > > **W2:** Dependence on BC support quality. Because actions are restricted to the BC support, any mis-estimated or overly conservative BC support could cap achievable performance; the paper could analyze this failure mode more explicitly (e.g., diagnostics when the BC model under-covers optimal modes)
> >
> > Thanks for bringing this up. We consider this as a **limitation**, instead of a weakness, of our work, along with **all** offline RL works that either **explicitly** learn a BC model or **implicitly** regularize the statistical distance between the learned policy and the BC policy.
> > We **have mentioned this in the limitation section** of our original submission and proposed potential solutions as future work:
> >
> > "Our method ensures that the policy $\pi_\theta$ remains within the support of the BC policy, meaning that it inherits any potential OOD errors made by the BC model itself; integrating behavior cloning methods with stricter support constraints could mitigate this dependence."
> >
> > The reviewer raised an insightful future direction: diagnostics when the BC model undercovers optimal modes. Although this is out of the scope of our paper, we **have added it in our limitation section**.

---

### Official Review · Reviewer_Q1Am · 2025-10-31

**Soundness:** 3
**Presentation:** 4
**Contribution:** 3
**Rating:** 4
**Confidence:** 4

**Summary:**

This paper presents ReFORM, a new method for on-support offline reinforcement learning.
The authors observe that existing approaches—such as CQL, IQL, and FQL—rely on explicit distance-based regularization (e.g., KL, Wasserstein, MMD) between the learned policy and the behavior policy to avoid out-of-distribution (OOD) actions.
However, these constraints depend heavily on tuning hyperparameters and can lead to excessive conservatism.

ReFORM instead enforces the support constraint by construction.
It introduces a two-stage reflected flow policy:

Behavior-Cloned Flow — learns a bounded latent-to-action mapping with uniform latent noise U(Bₗᵈ), guaranteeing that all generated actions lie within the dataset support.

Reflected Noise Flow — optimizes the latent noise inside the same bounded domain via reflection dynamics, maximizing Q values while preserving support inclusion.

The paper provides theoretical proofs that the reflected flow maintains support containment for both latent and action distributions (Theorems 1 & 2), eliminating OOD actions.
Empirically, on OGBench (40 tasks with clean and noisy datasets), ReFORM achieves the best average performance among FQL, IFQL, and DSRL—without hyperparameter tuning.

**Strengths:**

Originality:
ReFORM introduces a conceptually fresh idea—noise reflection for on-support control—that differs from prior regularization-based approaches.
The theoretical analysis clarifies when KL or Wasserstein constraints fail to guarantee support inclusion, which is insightful.

Technical Quality:
The paper provides rigorous derivations and clear theorems connecting reflection dynamics to support preservation.
The algorithmic design (bounded latent + reflected flow + distillation) is internally consistent and implementable.

Clarity :
Writing, figures, and notation are excellent.
The relationship between theory and algorithm is clearly explained, and the overall flow of the paper is easy to follow.

Significance:
The idea of enforcing support “by construction” could inspire future work in flow- or diffusion-based offline RL.
The “hyperparameter-free” advantage is appealing for practical use, though broader validation is needed.

**Weaknesses:**

Limited Empirical Scope: Experiments are confined to OGBench; no evaluation on D4RL, Adroit, or visual RL tasks.
The claim of “state-of-the-art performance” is weakened by the absence of comparison with recent strong baselines such as A2PR (arXiv 2405.19909), XQL, and EDQL.

Lacking computational cost analysis—reflection dynamics and distillation likely add overhead.

Moderate Empirical Depth: Ablation results (λ, number of clusters, reflection strength) are only in the appendix; some should appear in the main text.

It is unclear how the reflected flow interacts with other policy constraints (e.g., entropy regularization or behavior cloning loss).

While “support-by-construction” is elegant, it may restrict exploration in mixed-quality datasets.

**Questions:**

How does ReFORM perform compared with A2PR (Adaptive Advantage-Guided Policy Regularization) or XQL on D4RL benchmarks?
Including these baselines would help assess competitiveness against the latest SOTA.

What is the computational overhead of the reflection ODE and distillation step relative to FQL or IQL?

Could the reflection principle be combined with diffusion-based policies or used as a projection operator for general offline RL methods?

How sensitive is ReFORM to the latent domain size? Would a learned bound improve performance?

Can the authors release the full code to confirm reproducibility of OGBench results?

---

> ### Author Response · Authors · 2025-11-21
> **Author Reply (1/4)**
>
> We thank the reviewer for acknowledging our insightful theoretical analysis, rigorous derivation, clear theorems, clear writing, and the potential inspiration for future works.
>
> We hope our responses below and our revised manuscript address the concerns raised by the reviewer.
>
> ## Summary
>
> In brief, we have:
>
> - Provided additional experimental results on D4RL antmaze, D4RL adroit, and OGBench visual manipulation tasks, compared with [1, 2].
> - Provided details about the computational cost and discussed potential ways to improve it.
> - Explained the lack of exploration as a fundamental limitation of the offline RL problem setting.
> - Explained how the reflection principle can be combined with diffusion-based policies.
> - Pointed out that we have already provided experimental results on the sensitivity of ReFORM to the latent domain size.
> - Pointed out that we have already included the code in the supplementary material.
>
> However, the reviewer may have some misunderstandings of our paper. Specifically:
>
> - The reviewer mentioned ablation results about $\lambda$, number of clusters, and reflection strength. However, $\lambda$, number of clusters, and reflection strength **do not exist in our approach** as parameters or notations.
> - The reviewer mentioned that ablation results should be mentioned in the main text. However, almost all of the ablation results were in the main text of the original submission.
> - The reviewer mentioned "other policy constraints" including entropy regularization and behavior cloning loss, but we **do not consider entropy regularization** in our paper, and we are not sure how behavior cloning loss is considered as a policy constraint.
> - The reviewer mentioned a potential baseline EDQL, but we cannot find an algorithm with this name in the offline RL community.
>
> We tried our best to reply to the reviewer's concerns below, and would greatly appreciate it if the reviewer could clarify more about their concerns if we misunderstood anything!

---

> > ### Author Response · Authors · 2025-11-21
> > **Author Reply (2/4)**
> >
> > ## Detailed Reply
> >
> > > **W1/Q1:** Limited Empirical Scope: Experiments are confined to OGBench; no evaluation on D4RL, Adroit, or visual RL tasks. The claim of “state-of-the-art performance” is weakened by the absence of comparison with recent strong baselines such as A2PR (arXiv 2405.19909), XQL, and EDQL.
> >
> > Thanks for the suggestion. First, we **conducted additional experiments** on D4RL antmaze, D4RL adroit, and OGBench visual manipulation tasks:
> >
> > **D4RL antmaze (normalized return):**
> >
> > | Environment               | IFQL              | FQL (L)   | FQL (M)            | FQL (S)    | ReFORM             |
> > |---------------------------|-------------------|-----------|--------------------|------------|--------------------|
> > | antmaze-umaze-v2          | $91\pm7$          | $85\pm4$  | $\mathbf{99}\pm1$  | $88\pm13$  | $\mathbf{97}\pm0$  |
> > | antmaze-umaze-diverse-v2  | $55\pm28$         | $57\pm10$ | $\mathbf{88}\pm5$  | $61\pm26$  | $\mathbf{83}\pm3$  |
> > | antmaze-medium-play-v2    | $3\pm4$           | $14\pm6$  | $\mathbf{92}\pm1$  | $52\pm15$  | $85\pm4$           |
> > | antmaze-medium-diverse-v2 | $24\pm34$         | $9\pm4$   | $\mathbf{81}\pm13$ | $24\pm30$  | $\mathbf{80}\pm4$  |
> > | antmaze-large-play-v2     | $17\pm21$         | $43\pm10$ | $61\pm21$          | $3\pm4$    | $\mathbf{71}\pm4$  |
> > | antmaze-large-diverse-v2  | $28\pm27$         | $55\pm4$  | $\mathbf{85}\pm8$  | $8\pm12$   | $69\pm9$           |
> >
> > **D4RL adroit (normalized return):**
> >
> > | Environment               | IFQL              | FQL (L)   | FQL (M)            | FQL (S)    | ReFORM             |
> > |---------------------------|-------------------|-----------|--------------------|------------|--------------------|
> > | pen-human-v1              | $\mathbf{65}\pm1$ | $48\pm0$  | $59\pm4$           | $31\pm4$   | $\mathbf{64}\pm7$  |
> > | pen-cloned-v1             | $\mathbf{81}\pm8$ | $61\pm7$  | $66\pm5$           | $57\pm6$   | $70\pm6$           |
> > | pen-expert-v1             | $120\pm3$         | $105\pm7$ | $\mathbf{128}\pm1$ | $107\pm10$ | $\mathbf{129}\pm7$ |
> > | door-human-v1             | $3\pm1$           | $2\pm1$   | $0\pm0$            | $0\pm0$    | $\mathbf{4}\pm1$   |
> > | door-cloned-v1            | $-0\pm0$          | $0\pm0$   | $\mathbf{3}\pm2$   | $0\pm0$    | $1\pm1$            |
> > | door-expert-v1            | $89\pm5$          | $\mathbf{104}\pm1$ | $\mathbf{105}\pm0$ | $\mathbf{102}\pm0$ | $\mathbf{104}\pm4$ |
> >
> > **OGBench visual manipulation (return):**
> >
> > |Task|Dataset|IFQL|FQL(L)|FQL(M)|FQL(S)|ReFORM|
> > |--- |---    |--- |---   |---   |---   |---   |
> > |visual-cube-single-play-singletask-task1-v0|CLEAN| $-117\pm7$ | $-150\pm16$ | $\mathbf{-110}\pm9$ | $-138\pm19$ | $\mathbf{-108}\pm12$ |
> > |visual-cube-single-noisy-singletask-task1-v0|NOISY| $-95\pm2$  | $-176\pm10$ | $-103\pm2$          | $-57\pm3$   | $\mathbf{-52}\pm7$   |
> >
> > We find that **ReFORM with constant hyperparameters achieves similar or better performance** for these tasks. **We added these new results to Appendix C.4 of the revision**.
> >
> > Second, recall that our main claim is that ReFORM satisfies support constraints by construction, thus avoiding the need to hand-tune the weights of any statistical distance regularizations.
> > We chose baselines with the **same structure** as our method (e.g., flow policies, one-step distillation), but different **constraint methods** (e.g., regularization weights); **this is clarified in contributions of Section 1 revision**.
> > In contrast,
> > - A2PR [1] generates high-advantage actions for dataset augmentation
> > - XQL [2] introduces a $Q$-value learning framework that directly estimates the optimal soft-value functions in max-entropy RL
> >
> > with both of their contributions **orthogonal** to ours and thus not directly comparable.
> > We can integrate [1,2] with ReFORM by adding either A2PR data augmentation or XQL $Q$-value learning to the raw FQL backbone.
> > However, this is out of the scope of our paper, and we leave it to future work.
> > **We have cited these papers and added discussions in the limitation section of the revision**.

---

> > > ### Author Response · Authors · 2025-11-21
> > > **Author Reply (3/4)**
> > >
> > > Furthermore, note that A2PR and XQL were published in 2024 and 2023, respectively.
> > > With **all of our baselines published in 2025**, we politely argue that **our baselines are more recent**.
> > > However, we still provide the comparison between ReFORM and these two works below.
> > > ReFORM, again with constant hyperparameters, achieves better results than XQL and comparable results to A2PR with their **best hand-tuned hyperparameters**.
> > > The results of A2PR and XQL are from their corresponding papers.
> > >
> > > | Environment               | A2PR | XQL | ReFORM             |
> > > |---------------------------|------|-----|--------------------|
> > > | antmaze-umaze-v2          | $\mathbf{99}\pm2$  | $\mathbf{94}$ | $\mathbf{97}\pm0$  |
> > > | antmaze-umaze-diverse-v2  | $\mathbf{85}\pm4$  | $\mathbf{82}$ | $\mathbf{83}\pm3$  |
> > > | antmaze-medium-play-v2    | $\mathbf{86}\pm10$ | $76$ | $\mathbf{85}\pm4$           |
> > > | antmaze-medium-diverse-v2 | $\mathbf{86}\pm5$  | $74$ | $80\pm4$  |
> > > | antmaze-large-play-v2     | $\mathbf{71}\pm6$  | $47$ | $\mathbf{71}\pm4$  |
> > > | antmaze-large-diverse-v2  | $53\pm10$ | $49$ | $\mathbf{69}\pm9$           |
> > >
> > > Finally, to our best knowledge, **we cannot find an algorithm named "EDQL"** that is related to the offline RL community.
> > > We would greatly appreciate it if the reviewer could provide a citation to this work or clarify any misunderstandings.
> > > We will be glad to include any relevant citations in our paper.
> > >
> > >
> > > ---
> > >
> > > > **W2/Q2:** Lacking computational cost analysis - The computational overhead of the reflection ODE and distillation step relative to FQL or IQL.
> > >
> > > Great question.
> > > We argue that the reflection ODE does not add much additional computation because the reflected Euler method (12) has the same complexity as the Euler method (11), thanks to our choice of a hypersphere latent space.
> > > **We have added this in our revision after Equation (12).**
> > > The distillation step decreases computational cost by avoiding solving the flow ODE for the BC policy during inference.
> > >
> > > ---
> > >
> > > > **W5:** While “support-by-construction” is elegant, it may restrict exploration in mixed-quality datasets.
> > >
> > > Thanks for the comment. First, we want to emphasize that this paper considers the **offline** RL setting, where **no exploration is allowed** (mentioned in the first sentence of the abstract, the first sentence of Section 1, and the first paragraph of Section 3). Thus, the exploration restriction may be considered as a **limitation of the problem**, instead of a weakness of our method.
> > >
> > > Second, when mixed-quality datasets are used, our method is generally better than baselines because we enforce the support constraint, which is more relaxed than the statistical distance regularization considered in previous works.
> > > **We have demonstrated the high performance of ReFORM in low-quality datasets in our experiments** (see results with the noisy dataset and Figure 2 (right)).
> > >
> > > ---
> > >
> > > > **Q3:** Could the reflection principle be combined with diffusion-based policies or used as a projection operator for general offline RL methods?
> > >
> > > Good question! Our proposed reflected flow-based noise generator is a method for generating latent space actions, which can be combined with **any** generative models, including diffusion-based policies. We have included this in the Conclusion section of our revision.
> > >
> > > ---
> > >
> > > > **Q4:** How sensitive is ReFORM to the latent domain size? Would a learned bound improve performance?
> > >
> > > Thanks for the question! We **have already studied the sensitivity of ReFORM to the latent domain size** in Appendix C.4, paragraph "Ablations on the radius of the hypersphere $\mathcal B_l^d$". The results have been shown in Figure 17, which suggests that ReFORM is **not sensitive** at all to the latent domain size.
> > >
> > > ---
> > >
> > > > **Q5:** Can the authors release the full code to confirm reproducibility of OGBench results?
> > >
> > > We have already included the code for ReFORM in the supplementary materials and mentioned it in Appendix C.5 of the original PDF. We are working on creating a unified framework for ReFORM and all baselines and will release it in the final version.
> > >
> > > ---
> > >
> > > > **W3:** Ablation results ($\lambda$, number of clusters, reflection strength) are only in the appendix; some should appear in the main text.
> > >
> > > Thanks for the question. However, to our best knowledge, **there are no such aforementioned hyperparameters throughout our paper**. In addition, **all** our ablation studies, except for the latent domain size, are in the main text already. We did not put the latent domain size in the main text because ReFORM is not sensitive to it.
> > >
> > > Could the reviewer provide more information about what ablation results they referred to? We will be glad to explain more and put them in the main text.

---

> > > > ### Author Response · Authors · 2025-11-21
> > > > **Author Reply (4/4)**
> > > >
> > > > > **W4:** It is unclear how the reflected flow interacts with other policy constraints (e.g., entropy regularization or behavior cloning loss).
> > > >
> > > > Thanks for the question. However, we may not fully understand it. Is there any specific paper that is being referred to in this context?
> > > >
> > > > We provide some clarifications below, and we will really appreciate it if the reviewer can provide more explanations about the question.
> > > >
> > > > - "Other policy constraints": It is noteworthy that the only constraint we consider is the support constraint. There are no "other" policy constraints in our paper.
> > > > - "Entropy regularization": Our method does not consider entropy regularization, and our paper does not contain the word "entropy" at all.
> > > > - "Behavior cloning loss": This is the loss used for training the behavior cloning policy, which is not related to the reflected flow.
> > > >
> > > > Please let us know if we have misunderstood anything, as we would be happy to provide more explanation.
> > > >
> > > > ## References
> > > >
> > > > [1] Liu, Tenglong, et al. "Adaptive Advantage-Guided Policy Regularization for Offline Reinforcement Learning." ICML. 2024.
> > > >
> > > > [2] Garg, Divyansh, et al. "Extreme Q-Learning: MaxEnt RL without Entropy." ICLR. 2023.
> > > >
> > > > [3] Wagenmaker, Andrew, et al. "Steering your diffusion policy with latent space reinforcement learning". CoRL. 2025.

---

### Official Review · Reviewer_gwFF · 2025-11-01

**Soundness:** 3
**Presentation:** 3
**Contribution:** 3
**Rating:** 8
**Confidence:** 4

**Summary:**

This paper introduces an approach to offline RL that constrains the learned policy within the support of the data generating policy while still leveraging expressive policy classes.

**Strengths:**

The paper presents strong empirical results and thorough ablations justifying the design choices. In particular, ReFORM achieves strong performance across a variety of environments and tasks while using the same set hyperparameters, which is uncommon for offline RL algorithms.

**Weaknesses:**

The paper does not analyze potential reasons for why ReFORM outperforms the baselines in certain environments and datasets (clean vs noisy) but not others.

The paper does not investigate how this approach may scale to higher-dimensional state-action spaces. Does ReFORM’s approach of constraining to the data generating policy’s support work in a higher-dimensional space such as image-based inputs?

The paper does not compare to state-of-the-art algorithms on OGBench such as SORL [1] and floq [2]. (floq may constitute concurrent work and thus may not need to be compared against.)

The paper argues that ReFORM maintains high expressivity of the policy. However, the policy that is optimized via the Q-function is a one-step distillation policy, similar to FQL. One-step distillation policies have been shown to be less expressive than the multi-step policies employed by DSRL [3] and SORL [1].
The authors justify this design choice via an ablation (Figure 4), though these results seem to contradict [1]’s findings. Are there possible explanations for this?

- [1] Espinosa-Dice et al., “Scaling Offline RL via Efficient and Expressive Shortcut Models”
- [2] Agrawalla et al., “floq: Training Critics via Flow-Matching for Scaling Compute in Value-Based RL”
- [3] Wagenmaker et al., “Steering Your Diffusion Policy with Latent Space Reinforcement Learning”

**Questions:**

Did the authors consider using a Gaussian-based policy for noise generation (similar to DSRL) instead of the flow-based model? See other questions above.

---

> ### Author Response · Authors · 2025-11-21
> **Author Reply (1/2)**
>
> We thank the reviewer for recognizing our strong empirical results and thorough ablations, and for acknowledging our algorithm's strong performance across a variety of environments with the same set of hyperparameters.
>
> We hope our responses below and our revised manuscript address the concerns raised by the reviewer.
>
> ## Summary
>
> In brief, we have:
>
> - Provided new experimental results in the visual manipulation environments in OGBench.
> - Conducted a new ablation for studying a Gaussian-based policy for noise generation.
> - Clarified the main claim of our paper and our choice of baselines, and added discussions about how our method can potentially benefit from [1, 2].
> - Explained the performance decrease of ReFORM(NoDistill).
> - Discussed the performance variation of different methods on different tasks.
>
> The following detailed reply answers all weaknesses and questions raised by the reviewer, following the above order.
>
> ## Detailed Reply
>
> > **W2:** Does ReFORM’s approach of constraining to the data generating policy’s support work in a higher-dimensional space such as image-based inputs?
>
> Good question! To study ReFORM's performance in a higher-dimensional space, we **conducted additional experiments on visual manipulation environments** in OGBench. The return values are reported below:
>
> |Task|Dataset|IFQL|FQL(L)|FQL(M)|FQL(S)|ReFORM|
> |--- |---    |--- |---   |---   |---   |---   |
> |visual-cube-single-play-singletask-task1-v0|CLEAN| $-117\pm7$ | $-150\pm16$ | $\mathbf{-110}\pm9$ | $-138\pm19$ | $\mathbf{-108}\pm12$ |
> |visual-cube-single-noisy-singletask-task1-v0|NOISY| $-95\pm2$  | $-176\pm10$ | $-103\pm2$          | $-57\pm3$   | $\mathbf{-52}\pm7$   |
>
> Even on image-based inputs, **ReFORM, with constant hyperparameters, still outperforms all baselines even with tuned hyperparameters**.
> We did not compare against DSRL [3] because they did not include experiments in these environments, so DSRL's best hyperparameters are unknown.
>
> We have included these results in Appendix C.4 and Table 6 in the revision.
>
>
> ---
>
> > **Q1:** Did the authors consider using a Gaussian-based policy for noise generation (similar to DSRL) instead of the flow-based model?
>
> Good question! This can be a great additional ablation for answering Q4 (Section 5.3) in our paper. We **conducted a new experiment** by replacing ReFORM with a squashed Gaussian-based noise generator (similar to DSRL). The results are shown in Figure 4 (right) and discussed in paragraph Q4 in Section 5.3 in the revision. The results show that this variant (ReFORM(Gaussian)) performs worse than ReFORM because a Gaussian policy fails to capture multimodal noise distributions.
>
> ---
>
> > **W3:** The paper does not compare to state-of-the-art algorithms on OGBench such as SORL [1] and floq [2]. (floq may constitute concurrent work and thus may not need to be compared against.)
>
> Thanks for the suggestion! We would like to clarify that the main claim we are making is that ReFORM satisfies support constraints by construction, thereby omitting the requirement for hand-tuned weights of statistical distance regularizations. To support this claim, we choose baselines with the **same structure** (flow policies, one-step distillation) but different **policy regularization methods** (different regularization weights, different noise generating methods, etc). **We have made it clearer in the stated contributions (last paragraph of Section 1) in the revision.**
>
> On the other hand, SORL [1] applies shortcut models to replace the original flow model, and floq [2] trains the critic with flow matching. Their contributions are **orthogonal** to ours. Therefore, we did not include them in our experiments.
>
> Moreover, our approach can be easily **combined** with [1,2] by replacing the raw FQL backbone with SORL/floq, and can potentially benefit from them. However, this is out of the scope of this paper, and we leave it to future work. **We have cited these papers and added discussions in the limitation section of the revision.**
>
> We also agree with the reviewer that floq [2] should be considered as **concurrent** work because it was first submitted to Arxiv on Sep 8, 2025, while the ICLR abstract deadline is Sep 19, 2025, with only **10 days** difference.

---

> > ### Author Response · Authors · 2025-11-21
> > **Author Reply (2/2)**
> >
> > > **W4:** One-step distillation policies have been shown to be less expressive than the multi-step policies employed by DSRL [3] and SORL [1]. The authors justify this design choice via an ablation (Figure 4), though these results seem to contradict [1]’s findings. Are there possible explanations for this?
> >
> > Great question!
> >
> > First, note that our results from Figure 4 (right) are collected from the `cube-single` environment.
> > Looking at the `cube-single` results in Table 2 of [1], SORL (shortcut policy) performs **similarly** to FQL (one-step policy), suggesting that the expressiveness of a one-step policy is enough for solving **this** task.
> >
> > Second, we do not make any claim that ReFORM(NoDistill) is slightly worse than ReFORM because of the difference in **expressiveness**.
> > Instead, we believe that a long backpropagation chain may be **harmful** for training stability.
> > At least for this environment, the backpropagation issue seems to be the main reason why ReFORM(NoDistill) is slightly worse. This also matches the observation in [5]. (See Q6 in Section 5.3.)
> >
> > Finally, ReFORM can be easily combined with other distillation models like SORL [1], as we explained in reply to W3. We leave it to future work.
> >
> > ---
> >
> > > **W1:** The paper does not analyze potential reasons for why ReFORM outperforms the baselines in certain environments and datasets (clean vs noisy) but not others.
> >
> > Thanks for bringing this up!
> > This is a hard question to answer, as there can be many reasons, including stochasticity in training and testing, differing ground-truth $Q$-function landscapes, and more.
> >
> > The performance of offline RL algorithms generally varies in different environments and datasets, which **has also been observed in other popular papers** [1, 2, 3, 4, 5].
> > None of these papers analyzed the reasons why some methods outperform baselines in certain environments and datasets but not others, and it is still an open question.
> >
> > ## References
> >
> > [1] Espinosa-Dice et al., "Scaling Offline RL via Efficient and Expressive Shortcut Models"
> >
> > [2] Agrawalla et al., "floq: Training Critics via Flow-Matching for Scaling Compute in Value-Based RL"
> >
> > [3] Wagenmaker et al., "Steering Your Diffusion Policy with Latent Space Reinforcement Learning"
> >
> > [4] Park et al., "OGBench: Benchmarking Offline Goal-Conditioned RL"
> >
> > [5] Park et al., "Flow q-learning"

---

### Official Review · Reviewer_6RET · 2025-11-01

**Soundness:** 3
**Presentation:** 3
**Contribution:** 3
**Rating:** 6
**Confidence:** 4

**Summary:**

This manuscript proposes the ReForm Framework, to cope with the OOD problem in offline RL as well as the multimodal issue of optimal policy distribution. The method first proposes the method to map the source distribution to the supported action distribution, and then utilizes the generated bounded noise via Reflected Flow to optimize the Q value. Experiments on more than 40 challenging tasks demonstrate the superoirty of the proposed framework.

**Strengths:**

* The OOD issue, as well as the distribution of optimal action policy, are classic topics in offline RL, it is appreciated that the authors consider these issues from the new perspectives.

* The use of bounded source distribution and reflect flow is quite novel and appealing. It fundamentally avoids OOD actions being explored.

* The reflected flow noise generator can produce complex multimodal noise, which is helpful for some scenarios where real actions distribution are quite complex.

**Weaknesses:**

1. Some related references are missing, and it is suggested to consider the related work in the manuscript.

* https://arxiv.org/abs/2202.06239

* https://arxiv.org/abs/1705.08868

* https://arxiv.org/abs/2301.12130

2. The model design is appealing, however, the performance of the model also relies on the quality of behavior cloning model. How is the model performance if the BC model is not well estimated. Does the author consider about the robustness of the proposed method?

3. Although the authors suggest distillation model, the training cost still seems high. Does the author consider about this? Or it is suggested to consider some ablation study on this issue.

4. From the results in Figure 4, why ReForm(NoDistill) is slightly worse than ReForm, the reviewer is confused about this. For other variants, it seems their performances are much inferior to ReForm and ReForm(NoDistill). So is there a light and efficient version of ReForm?

**Questions:**

See the weakness above

---

> ### Author Response · Authors · 2025-11-21
> **Author Reply (1/2)**
>
> We thank the reviewer for recognizing us considering the OOD issue from the new perspectives, acknowledging the novelty of our algorithm as a fundamental solution for addressing the OOD problem, and acknowledging the ability of the reflected flow noise generator for producing multimodal noise.
>
> We hope that our responses below and our revised manuscript address the concerns raised by the reviewer.
>
> ## Summary
>
> In brief, we have:
>
> 1. Cited more related works.
> 2. Clarified the dependence on the BC model as a limitation of our work along with all offline RL works.
> 3. Provided details about the training cost and discussed potential ways to improve it in Appendix C.4.
> 4. Explained again the potential reason why ReFORM(Distill) is slightly worse than ReFORM in the ablation studies.
> 5. Discussed a potential lighter version of ReFORM.
>
> The following detailed reply answers all weaknesses and questions raised by the reviewer, following the above order.
>
> ## Detailed Reply
>
> > **W1:** Some related references are missing, and it is suggested to consider the related work in the manuscript.
>
> Thanks for the suggestion! We have included the related works [1,2,3] suggested by the reviewer in the revision.
>
> - [1]: Cited in paragraph 1 of Section 2.
> - [2]: Cited in the paragraph after Equation (5).
> - [3]: Cited in paragraph 1 of Section 2 and the paragraph after Equation (5)
>
> ---
>
> > **W2:** The performance of the model also relies on the quality of behavior cloning model. Does the author consider about the robustness of the proposed method?
>
> Thanks for bringing this up.
> We consider the dependence on the BC model as a **limitation**, not a weakness, of our work.
> Note that we share this same limitation with **all** offline RL works that either **explicitly** learn a BC model or **implicitly** regularize the statistical distance between the learned policy and the BC policy.
> We **mentioned this in the limitation section** of our original submission and proposed potential solutions as future work:
>
> _"Our method ensures that the policy $\\pi\_\\theta$ remains within the support of the BC policy, meaning that it inherits any potential OOD errors made by the BC model itself; integrating behavior cloning methods with stricter support constraints could mitigate this dependence."_
>
> We have also **proposed additional potential solutions** in the limitation section of our revision, including diagnosing when the BC model generates OOD errors, or applying a pre-trained BC model.
> However, they are out of the scope of this work, and we leave them to future work.
>
>
> ---
>
> > **W3:** Although the authors suggest distillation model, the training cost still seems high.
>
> Thanks for bringing this up! We have already mentioned the computational complexity as a **limitation** of our work in Section 6.
>
> Although we have reported the training time for ReFORM in Appendix C.1, we report the training time for all baselines below for better comparison:
>
> |Algorithm|ReFORM|FQL|IFQL|DSRL|
> |---------|------|---|----|----|
> |Training time (minutes, $10^6$ steps)|80|40|35|55|
>
> The table shows that ReFORM indeed doubles the training time compared to FQL due to the 2-stage flow. However, as shown in our experiments, FQL is sensitive to hyperparameters, and **searching for optimal hyperparameters requires significantly more runs**. And, the time advantage of IFQL comes at the expense of **poor performance** across the tasks, evident from the results in Table 4. On the contrary, ReFORM can be used without any hyperparameter searching.
>
> The training cost issue can also be mitigated by applying a pre-trained BC model and latent space RL, like DSRL [4]. We have **provided new discussions** on this in the limitation section and Appendix C.4 of our revision.
>
> ---
>
> > **W4 (1/2):** From the results in Figure 4, why ReForm(NoDistill) is slightly worse than ReForm?
>
> Thanks for the question. We have discussed this in paragraph Q6, Section 5.3 of our original submission. In short, a longer backpropagation chain can be harmful, which matches the observation in [5].
>
> ---
>
> > **W4 (2/2):** For other variants, it seems their performances are much inferior to ReForm and ReForm(NoDistill). So is there a light and efficient version of ReForm?
>
> Good question! Reviewer $\b$ also brought this up. For fair comparison, we implemented ReFORM with the raw FQL [5] backbone, but the FQL backbone itself can be potentially improved by shortcut models [6] or better $Q$-function learning approaches [7]. We have **added more discussions** in the limitation section of our revision.

---

> ### Author Response · Authors · 2025-11-21
> **Author Reply (2/2)**
>
> ## References
>
> [1] Wu, Jialong, et al. "Supported policy optimization for offline reinforcement learning." Advances in Neural Information Processing Systems 35 (2022): 31278-31291.
>
> [2] Grover, Aditya, Manik Dhar, and Stefano Ermon. "Flow-gan: Combining maximum likelihood and adversarial learning in generative models." Proceedings of the AAAI conference on artificial intelligence. Vol. 32. No. 1. 2018.
>
> [3] Zhang, Jing, et al. "Constrained policy optimization with explicit behavior density for offline reinforcement learning." Advances in Neural Information Processing Systems 36 (2023): 5616-5630.
>
> [4] Andrew Wagenmaker, Mitsuhiko Nakamoto, Yunchu Zhang, Seohong Park, Waleed Yagoub, Anusha Nagabandi, Abhishek Gupta, and Sergey Levine. Steering your diffusion policy with latent space reinforcement learning. arXiv preprint arXiv:2506.15799, 2025.
>
> [5] Seohong Park, Qiyang Li, and Sergey Levine. Flow q-learning. In International Conference on Machine Learning (ICML), 2025b.
>
> [6] Nicolas Espinosa-Dice, Yiyi Zhang, Yiding Chen, Bradley Guo, Owen Oertell, Gokul Swamy, Kianté Brantley, and Wen Sun. Scaling offline RL via efficient and expressive shortcut models. In The Thirty-ninth Annual Conference on Neural Information Processing Systems, 2025.
>
> [7] Bhavya Agrawalla, Michal Nauman, Khush Agrawal, and Aviral Kumar. floq: Training critics via flow-matching for scaling compute in value-based rl. arXiv preprint arXiv:2509.06863, 2025.

---

### Author Response · Authors · 2025-11-21
**Author Reply to All (1/2)**

$\def\a{\color{#648FFF}{\textsf{6RET}}} \def\b{\color{#E69F00}{\textsf{gwFF}}} \def\c{\color{#DC267F}{\textsf{Q1Am}}} \def\d{\color{#0FA83C}{\textsf{NkRQ}}}$

We thank the reviewers for their valuable comments.
We are excited that the reviewers have identified the novelty and efficacy of our idea ($\a$, $\c$, $\d$),
acknowledged the rigorous derivations and clear theorems ($\c$, $\d$), appreciated our "hyperparameter-free" advantage ($\b$, $\c$, $\d$),
strong empirical results and thorough ablations ($\b$, $\d$), and
good presentation (**all** reviewers).
We believe that ReFORM takes a significant step towards **completely avoiding the out-of-distribution (OOD) issue of offline RL without additional regularization weights while preserving the expressive multimodal policy**.

---

The primary concerns include testing on more benchmarks and higher-dimensional spaces ($\b$, $\c$, $\d$), analyzing the computational cost ($\a$, $\c$), and comparing with more baselines ($\b$, $\c$).

In our revision, we provide additional experimental results and discussions to clarify **all** raised concerns. Notable changes are  marked in $\color{#D80000}{\text{red}}$ in the revision.
We provide a brief summary below.

## 1. New experiments

In addition to the $40$ OGBench tasks presented already in our paper, we evaluate our method on the D4RL benchmark and OGBench visual manipulation tasks. The results are provided below. Even with image-based inputs and completely different benchmarks, ReFORM, with **constant hyperparameters**, still performs similarly with or better than all baselines even with tuned hyperparameters.

**D4RL antmaze (normalized return):**

| Environment               | IFQL              | FQL (L)   | FQL (M)            | FQL (S)    | ReFORM             |
|---------------------------|-------------------|-----------|--------------------|------------|--------------------|
| antmaze-umaze-v2          | $91\pm7$          | $85\pm4$  | $\mathbf{99}\pm1$  | $88\pm13$  | $\mathbf{97}\pm0$  |
| antmaze-umaze-diverse-v2  | $55\pm28$         | $57\pm10$ | $\mathbf{88}\pm5$  | $61\pm26$  | $\mathbf{83}\pm3$  |
| antmaze-medium-play-v2    | $3\pm4$           | $14\pm6$  | $\mathbf{92}\pm1$  | $52\pm15$  | $85\pm4$           |
| antmaze-medium-diverse-v2 | $24\pm34$         | $9\pm4$   | $\mathbf{81}\pm13$ | $24\pm30$  | $\mathbf{80}\pm4$  |
| antmaze-large-play-v2     | $17\pm21$         | $43\pm10$ | $61\pm21$          | $3\pm4$    | $\mathbf{71}\pm4$  |
| antmaze-large-diverse-v2  | $28\pm27$         | $55\pm4$  | $\mathbf{85}\pm8$  | $8\pm12$   | $69\pm9$           |

**D4RL adroit (normalized return):**

| Environment               | IFQL              | FQL (L)   | FQL (M)            | FQL (S)    | ReFORM             |
|---------------------------|-------------------|-----------|--------------------|------------|--------------------|
| pen-human-v1              | $\mathbf{65}\pm1$ | $48\pm0$  | $59\pm4$           | $31\pm4$   | $\mathbf{64}\pm7$  |
| pen-cloned-v1             | $\mathbf{81}\pm8$ | $61\pm7$  | $66\pm5$           | $57\pm6$   | $70\pm6$           |
| pen-expert-v1             | $120\pm3$         | $105\pm7$ | $\mathbf{128}\pm1$ | $107\pm10$ | $\mathbf{129}\pm7$ |
| door-human-v1             | $3\pm1$           | $2\pm1$   | $0\pm0$            | $0\pm0$    | $\mathbf{4}\pm1$   |
| door-cloned-v1            | $-0\pm0$          | $0\pm0$   | $\mathbf{3}\pm2$   | $0\pm0$    | $1\pm1$            |
| door-expert-v1            | $89\pm5$          | $\mathbf{104}\pm1$ | $\mathbf{105}\pm0$ | $\mathbf{102}\pm0$ | $\mathbf{104}\pm4$ |

**OGBench visual manipulation results (return):**

|Task|Dataset|IFQL|FQL(L)|FQL(M)|FQL(S)|ReFORM|
|--- |---    |--- |---   |---   |---   |---   |
|visual-cube-single-play-singletask-task1-v0|CLEAN| $-117\pm7$ | $-150\pm16$ | $\mathbf{-110}\pm9$ | $-138\pm19$ | $\mathbf{-108}\pm12$ |
|visual-cube-single-noisy-singletask-task1-v0|NOISY| $-95\pm2$  | $-176\pm10$ | $-103\pm2$          | $-57\pm3$   | $\mathbf{-52}\pm7$   |

---

> ### Author Response · Authors · 2025-11-21
> **Author Reply to All (2/2)**
>
> ## 2. Computational cost
>
> We have already mentioned the computational complexity as a **limitation** of our work in Section 6.
> We additionally provide the training cost (Table 7 of revision), and show that ReFORM indeed doubles the training time compared to FQL due to the 2-stage flow. However, as shown in our experiments, FQL is sensitive to hyperparameters, and **searching for optimal hyperparameters in practice requires significantly more runs**. On the contrary, **ReFORM can be used without any hyperparameter searching**.
>
> The training cost issue can also be mitigated by applying a pre-trained BC model and latent space RL, like DSRL [1]. We have **provided new discussions** on this in the limitation section and Appendix C.4 of our revision.
>
> ## 3. More baselines
>
> Recall that the main contribution of ReFORM is that it satisfies support constraints by construction, thus avoiding the need to hand-tune the weights for any statistical distance regularizations.
> To support this claim, we choose baselines with the **same structure** (flow policies, one-step distillation) but different **policy regularization methods** (different regularization weights, different noise generating methods, etc). **We have made it clearer in the stated contributions (last paragraph of Section 1) in the revision.**
>
> The related works that the reviewers mentioned have contributions **orthogonal** to ours. Therefore, we did not include them in our experiments.
> Moreover, our approach can be easily **combined** with these works by replacing the raw FQL backbone with these works. However, this is out of the scope of this paper, and we leave it to future work. **We have cited these papers and added discussions in the limitation section of the revision.**
>
> ---
>
> We hope the new results and clarifications will make our paper stronger.
> We tried our best to resolve **all** raised questions in the individual responses below. If the reviewers have any additional questions/comments/concerns, please let us know. We appreciate the reviewer's precious time in providing their valuable feedback!
>
> ## References
>
> [1] Andrew Wagenmaker, Mitsuhiko Nakamoto, Yunchu Zhang, Seohong Park, Waleed Yagoub, Anusha Nagabandi, Abhishek Gupta, and Sergey Levine. Steering your diffusion policy with latent space reinforcement learning. arXiv preprint arXiv:2506.15799, 2025.

---

### Meta-Review · Area_Chair_VWjX · 2026-01-06

**Summary:**

This paper proposes ReFORM, an offline RL framework that enforces on-support policy learning “by construction” rather than via explicit statistical-distance regularization. The core idea is to (i) learn a bounded behavior-cloned flow mapping from a bounded latent source to actions, ensuring generated actions remain within the behavior support, and (ii) optimize performance by updating latent noise through a reflected flow that preserves the bounded domain while enabling expressive (potentially multimodal) policies. Reviewers generally found the idea novel and the theoretical framing around support containment compelling, and they highlighted strong empirical performance (notably on OGBench with fixed hyperparameters).
The concerns that most influenced my decision revolve around:
Empirical scope and comparability: initial evaluations were largely confined to OGBench and a narrower set of baselines; multiple reviewers requested broader benchmarks (e.g., D4RL AntMaze/Adroit, visual tasks) and/or comparisons to stronger or more standard baselines.
Compute/training cost: the two-stage flow and distillation raise questions about runtime and practicality.
Dependence on BC quality / support estimation: restricting actions to BC support can inherit BC model errors or under-coverage, potentially capping performance; reviewers asked for robustness/failure-mode analysis.
Mechanistic understanding / scaling: why the method helps in some datasets/environments but not others, how it scales to higher-dimensional inputs (e.g., images), and how the “expressive policy” claim relates to using one-step distillation.
Overall, the rebuttal substantively addressed several major empirical-scope and practicality questions by adding new experiments (D4RL/visual) and cost reporting, which increases confidence; remaining concerns primarily relate to comparability to broader SOTA and BC-dependence analysis.

**Reviewer Concerns:**

Missing related work / citations (Reviewer 6RET): Authors added the suggested references and clarified positioning.
Lack of broader benchmark coverage (Reviewers Q1Am, NkRQ, gwFF): Authors added results on D4RL AntMaze, D4RL Adroit, and OGBench visual manipulation tasks, directly responding to the “only OGBench” concern and partially addressing scaling to higher-dimensional inputs.
Compute cost / overhead (Reviewers 6RET, Q1Am): Authors provided additional training-time comparisons and argued where the overhead comes from (two-stage flow) and how distillation helps at inference; they also discussed potential mitigation strategies.

**Reviewer Scores:**

Estimated score changes if reviewers had been able to fully participate in discussion (based on how directly their concerns were addressed):
Reviewer 6RET (initial: 6): likely 6 → 7. Missing citations were fixed; compute cost was quantified; confusion about distillation variants was addressed; limitations were clarified. Remaining BC-dependence concerns likely prevent a larger jump.
Reviewer gwFF (initial: 8): likely 8 → 8 (or 8 → 9 if very persuaded by the new visual results + Gaussian ablation). Their main requests (scaling-to-visual and Gaussian ablation) were addressed; SORL/floq comparisons were not done but were discussed as orthogonal.
Reviewer Q1Am (initial: 4): likely 4 → 5 (possibly 6 if they accept the authors’ framing of “orthogonal” baselines and are satisfied with the added D4RL/visual results and cost discussion). The rebuttal addresses the largest empirical-scope and compute concerns, but the baseline-comparability question may still limit how far they would raise.
Reviewer NkRQ (initial: 6): likely 6 → 6/7. The added D4RL/visual experiments address their main comparability concern. BC-support dependence remains, so I expect at most a modest increase.

---

### Decision · Program_Chairs · 2026-01-26

Accept (Poster)